# Slope Stability of a Scree Slope Based on Integrated Characterisation and Monitoring

**Daisy Lucas** [1,*] [ORCID]**, Kerstin Fankhauser** [2]**, Hansruedi Maurer** [2]**, Brian McArdell** [3]**, Reto Grob** [1,4]**, Ralf Herzog** [1]**, Ernst Bleiker** [1] **and Sarah M. Springman** [1]

[1] Institute for Geotechnical Engineering, ETH Zürich, Stefano-Franscini-Platz 5, 8093 Zürich, Switzerland; reto.grob@flumgeo.ch (R.G.); ralf.herzog@igt.baug.ethz.ch (R.H.); ernst.bleiker@igt.baug.ethz.ch (E.B.); sarah.springman@sl.ethz.ch (S.M.S.)

[2] Institute for Geophysics, ETH Zürich, Sonneggstrasse 5, 8092 Zürich, Switzerland; kerstin.fankhauser@erdw.ethz.ch (K.F.); hansruedi.maurer@erdw.ethz.ch (H.M.)

[3] Unit Research Mountain Hydrology and Mass Movements, Swiss Federal Research Institute WSL, Zürcherstrasse 111, 8903 Birmensdorf, Switzerland; brian.mcardell@wsl.ch

[4] Geotechnical Engineering/Geology, FlumGeo AG, Fuchsenstrasse 19, St. 9016 Gallen, Switzerland

* Correspondence: daisy.lucas@igt.baug.ethz.ch; Tel.: +41-44-633-3430

**Abstract:** Three years of geotechnical seasonal field monitoring including soil temperature, suction and volumetric water content plus geophysical measurements, lead to a preliminary ground model and assessment of slope stability for a steep scree slope in the Meretschibach catchment, near Agarn village in the Swiss Alps. Building on data reported in a previous paper, which focused on preliminary ground characterisation and seasonal field monitoring, this current research aims to understand whether a surficial failure in the scree slope, triggered by rainfall and depending on bedrock conditions, would represent a relevant natural hazard for Agarn village. A final year of field data is included as well as site-specific sensor calibration, a Ground Penetrating Radar (GPR) profile, and laboratory triaxial testing to provide strength parameters. A bedrock map is presented, based on GPR, with a realistic ground model of the entire scree slope. Furthermore, a preliminary numerical analysis, performed using SEEP-SLOPE/W, shows the influence of a bedrock outcrop observed in the field, for a specific soil thickness, strength parameters and rain intensity. The stability of a gravelly slope decreases with groundwater flow over a step in the bedrock, and the location of the failure will tend to move uphill of a bedrock outcrop at a shallow depth as groundwater flow increases.

**Keywords:** landslide; monitoring; bedrock; scree slope; Ground Penetrating Radar; volumetric water content; natural hazard; triaxial stress path testing

## 1. Introduction

Scree slopes are formed in mountain areas by rock debris, which loses potential energy in falling from weathered and fractured bedrock walls, and follows a downslope trajectory until it decelerates and stops, i.e., to zero kinetic energy once again. Typical dynamic processes occurring in an active scree slope are toppling failures [1], rockfalls; landslides (small or large mass movements); and the fragmenting and sorting of materials [2–4].

The field site is located between 1840–1910 m.a.s.l. in a mountain slope of the Meretschibach catchment in canton Valais, Switzerland (Figure 1a). Agarn village has approximately 800 inhabitants and lies in the Rhone river valley at 637 m.a.s.l. The catchment area of about 9.2 km$^2$ is divided into four subcatchments [5], from which Bochtür is observed as the most active debris channel in the area [5], and it is also the most active in terms of rockfalls and debris flow events. It poses a persistent hazard for the Agarn community.

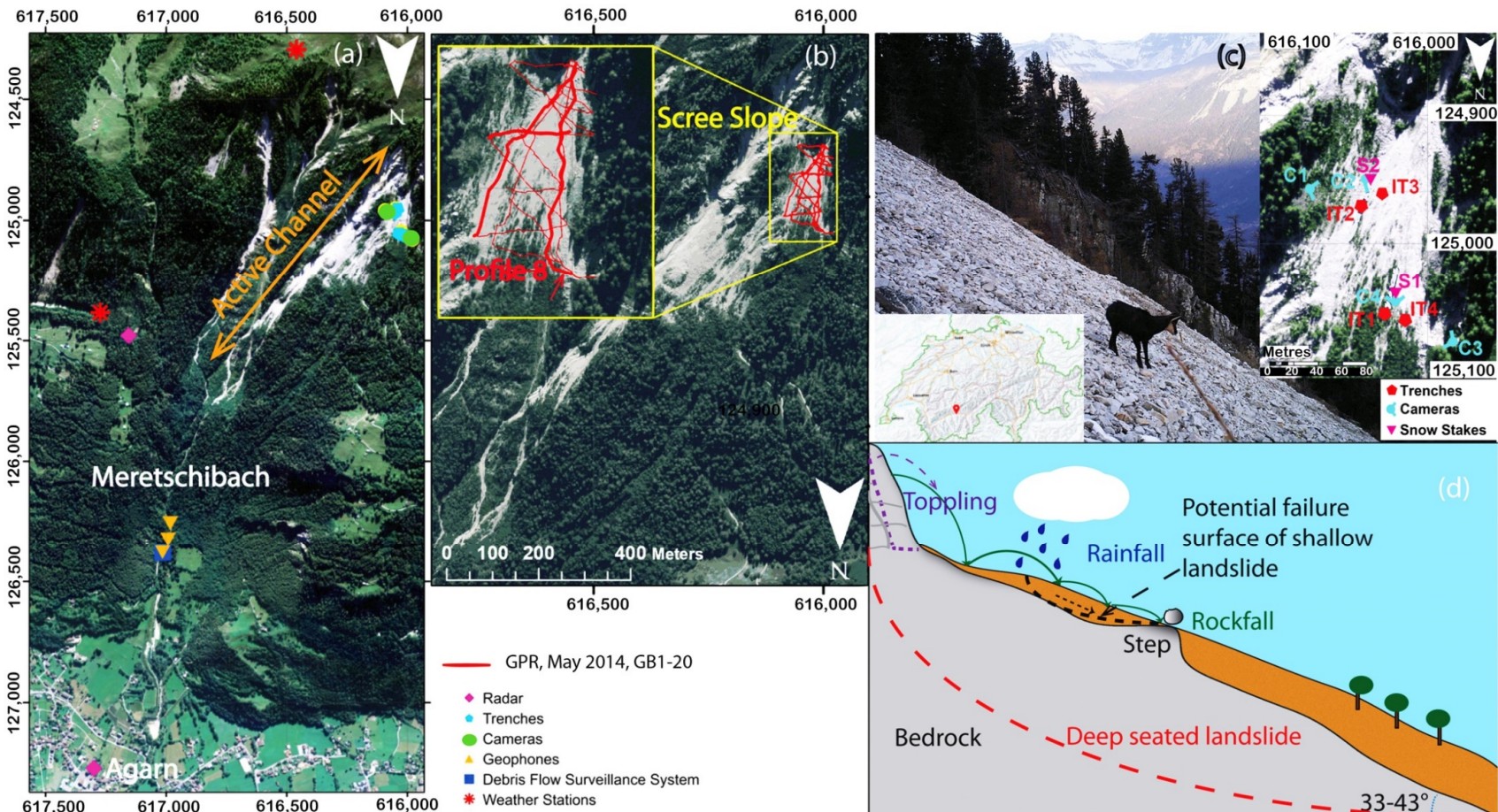

**Figure 1.** Field site located in canton Valais, Switzerland. (**a**) Location of geotechnical and remote sensing, (**b**) Geophysical measurements, extent of GPR with profile 8 indicated. Distances according to Swiss coordinate system CH1903 (in metres), (**c**) overview of the investigated area, (**d**) mass movement instabilities in the scree slope. The likely presence of a deep-seated landslide is indicated schematically with a red dashed line.

The bedrock slope lies on the west side of the Bochtür and is inclined mainly between 33°–43° [6]. It is the selected location for the geotechnical characterisation. The heavily jointed quartzite bedrock emerges at higher elevations as outcrops that are exposed to strong mechanical weathering, producing boulders of various sizes for rockfalls and toppling events. These impact on the slope and bounce, roll and fragment until they are deposited prior to subsequent remobilisation along the scree slope, covering the underlying layer of gravelly soil. Geological observations suggest that the bedrock is stepped, with joints dipping southwards (into the slope) with an inclination of 10°–30° [7,8].

The scree slope at Meretschibach (Figure 1a) consists of a surficial layer of large boulders and cobbles underlain by a predominantly gravelly soil layer, with a mix of coarser, larger and finer sizes within a heterogeneous structure. The characteristics, in terms of geometry and particle arrangement, depend on the energy balances and rock crushing strength (e.g., in extremis, as seen in rock slides and rock avalanches [9–16]). This implies that particles of all sizes, and with fractal dimensions, are created during fragmentation, leading to a fabric where larger particles are immersed in a matrix of finer ones [11,14,17–19]. Furthermore, if the input energy is high, the rock may experience rock-burst [20], where highly stressed rock disintegrates suddenly, reaching high velocities (>10 m/s) [13]. Additional factors are the bedrock slope and the climate environment.

Scree slopes that are under dynamic processes of debris deposition and remobilization, may suffer instabilities under hydrological events through intense periods of rainfall, or snow melting processes. Scree slope formation, processes and dynamics have been documented in the literature [2–4,21–24]; although there is a lack of data on the slope instability and failure mechanisms.

A scree slope in the Meretschibach catchment in canton Valais in the south of Switzerland (Figure 1a) was characterised in order to improve understanding of the process phenomena in scree slopes and to investigate the potential hazard following initiation of landslides/debris flow with runout on to Agarn village. Frequent toppling and rockfall events were observed in the scree slope (Figure 1b), but their volume was too small (order of magnitude) to be considered as a hazard to the village. The most severe hazard that could be predicted after three years of seasonal field monitoring, apart from a deep-seated landslide, which was not investigated in this research, was identified as a surficial landslide. This could mobilise a significant volume of debris that could eventually be deposited in an active channel and subsequently turn into a debris flow that could reach the community in Agarn.

Preliminary results from this research were presented in the paper. 'Application of geotechnical and geophysical field measurements in an active alpine environment' [6], delivering two years of data of volumetric water content (VWC), temperature records at specific locations in the slope (Figure 1c), depth to bedrock (downslope) and characterisation of the gravelly soil. The data led to a better understanding of the response of the surficial soil in a scree slope, due to annual variations in weather conditions in an alpine environment.

This paper adds one more year of data to that already published, providing a complete overview of the seasonal field monitoring conducted at Meretschibach. Additionally, a soil-specific calibration was carried out for the VWC estimation in terms of temperature variation, accompanied by further analysis of GPR data (Figure 1b) and the shear strength analysis derived from stress-path triaxial testing of the gravelly soil.

The VWC distribution at a depth of 1 m [6], showed for a period of at least two weeks, that a thin layer of soil in one location was near saturation, after successive rainfall events at the beginning of this period. Had there been additional severe events in the following fortnight, a surficial landslide could have occurred due to increase in thickness of the saturated zone and further reduction in effective stress. Failure could be anticipated, should this change in saturation extend over a larger area of the slope (Figure 1d). In this scenario, some of the mobilised soil debris could be contained by the forest downslope, and the rest could fall into the active channel, accumulating there, and increasing the hazard of a debris flow in the future.

The results of the soil-specific calibration for temperature variation in the positive range showed that the VWC measurements at the different locations are more influenced by the heterogeneity of the

silty gravel than by temperature changes. Application of the GPR led to a map of bedrock depth of scree slope, including steps in the profile. Subsequently, an estimation was made of the slope stability through simplified numerical models, based on strength parameters obtained by the triaxial stress path tests.

## 2. Methodology

### 2.1. Preliminary Hazard Assessment

Initially, a preliminary hazard assessment was made. The Bochtür hillslope is well known to local land-use planners for persistent rockfall and debris flow activity. Observations of the movement of the hillslope described herein are based on unpublished project reports from the Swiss Federal Institute for Forest, Snow and Landscape Research (WSL), summarised in McArdell [25].

InSAR measurements of displacement suggest that part of the Bochtür hillslope is underlain by a large landslide, with the head of the landslide corresponding roughly to the unvegetated areas at the uppermost part of the Bochtür hillslope and extending about 1.25 km downslope into the forest, with a width of up to about 0.5 km [26]. Photogrammetric analyses of the unvegetated portion of the Bochtür, made by WSL staff for the period between 1959 and 2010, suggest a somewhat incoherent movement pattern, with many areas exhibiting surficial movements of less than 0.1 m/year.

Locally large velocities (up to 0.4 m/year) have been measured, however it is unclear whether they represent surface creep on the talus, toppling movements of bedrock steps described above, or local differential landslide deformation. Differential Global Positioning System (GPS) measurements of marked boulders on the surface of the slope (Figure 2a), made between 28 October 2013 and 30 May 2014 (e.g., the period when the slope was relatively wet), indicate annual movement rates of up to 0.4 m/year (Figure 2b). The presence of Sackung trench-scarp features indicated on the geological map [7,8] in the forest adjacent to the Bochtür, and in the vegetated area at the head of the Bochtür (Figure 3a), suggests that creep deformation is occurring within the bedrock. However, no additional information is available from borehole data or geophysical measurements to indicate the depth of a possible failure plane or planes (Figure 3b). Rock avalanche deposits can be observed in the adjacent Illgraben catchment to the west, and the hillslope to the east of the Bochtür is partially covered by large blocky debris, indicating that the possibility of rapid large-scale landslides should not be ignored at this site.

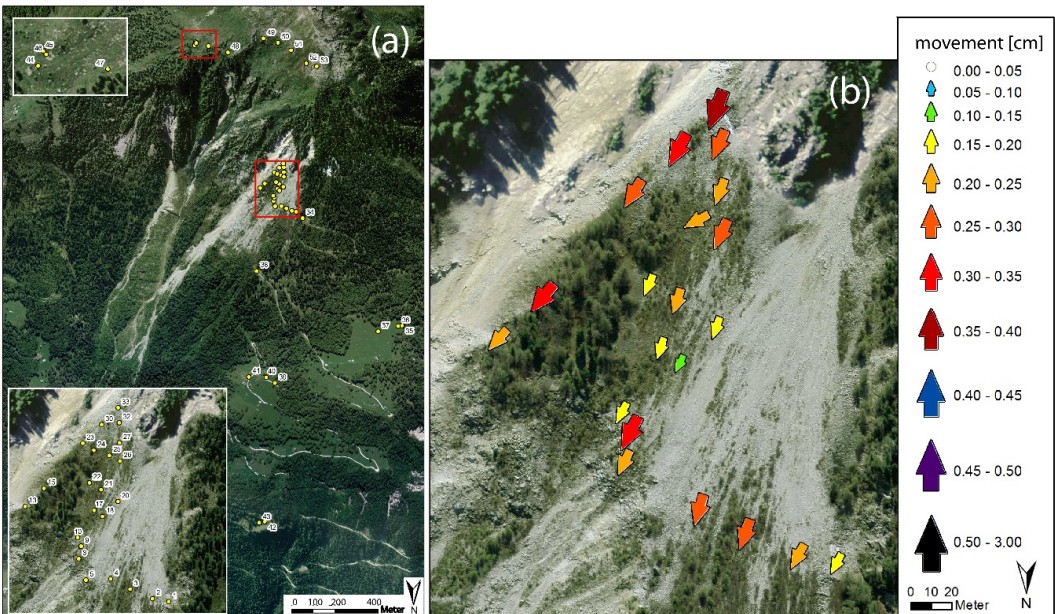

**Figure 2.** The Meretschibach-Böchtur catchment. (**a**) Location of GPS points and (**b**) the annual hillslope movement rate between 28 October 2013 and 30 May 2014. From Oggier and Thee [27].

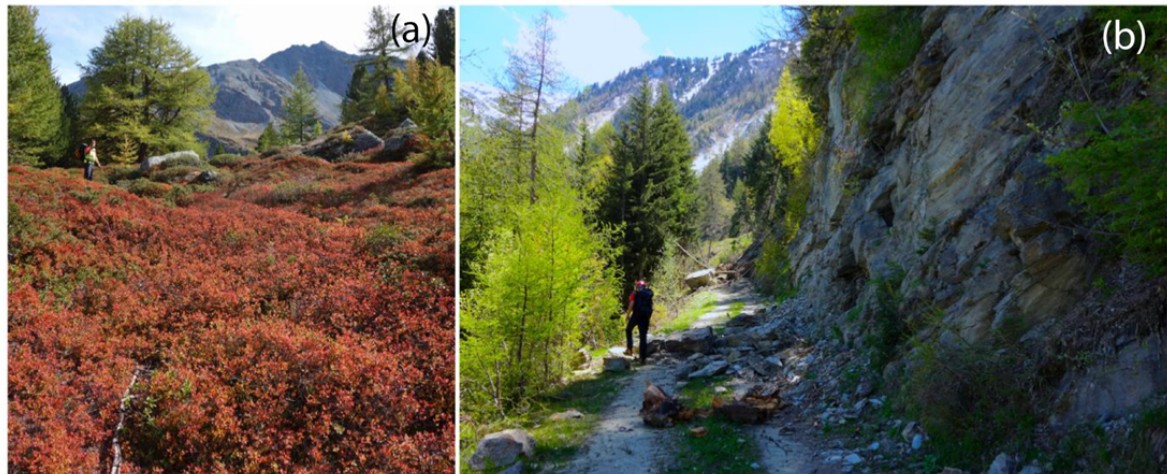

**Figure 3.** (**a**) Closed elongated depression at the top of the Böchtur hillslope, possibly indicating a Sackung-type failure of the bedrock, (**b**) local slope collapses near the bend of the road described in the text. Photos by B. McArdell, WSL.

Six debris flows were observed in the Meretschibach torrent between July 2014 and September 2015, during rainstorms with durations of less than one day to up to 5 days. It is not appropriate, in a statistical sense, to construct a rainfall intensity-duration threshold (ID) curve for only six debris flow events. However, it is worth noting that all of the points plot above the triggering threshold for the Illgraben catchment [28] and all plot lower than a general Swiss ID curve established by Zimmermann [29].

Rockfall activity at the Bochtür and the interaction with the forest was investigated by Eichenberger [30] along four transects through the forest downslope of the Bochtür, with observations on more than 1500 trees. They described both fresh rockfall deposits (individual stones and boulders) and dated the age of rockfall wounds on the trees. Boulders in the forest with a clear rockfall origin have a median volume of ~0.1 $m^3$ with boulders of volumes up to 10 $m^3$ present (n = 204 boulders measured along the transects). Although not all stones and boulders impacted trees and caused damage to them, a large number of rockfall scars could be dated, and most of them were less than 10 years old, and were concentrated, as would be expected at the upslope border of the forest.

*2.2. Soil Characterisation*

Electrical resistivity tomography (ERT) results from geophysical surveys, soil classification tests and field observations from instrumented trenches (IT) enabled a preliminary ground model to be presented in Lucas et al. [6], which forms the basis of this study. The soil is classified according to the Swiss standard classification (SN 670 004-2NA) as poorly-graded gravel with silt and sand (GP-GM), silty gravel (GM), well-graded gravel with silt and sand (GW-GM), and well-graded gravel (GW). The particles are derived mostly from the quartzite bedrock outcrop above the slope, which is heavily jointed and highly susceptible to weathering, leading to rockfalls, sliding, and toppling [5,31].

ERT measurements performed by Fankhauser [31] and published in Lucas et al. [6] were used primarily to establish the depth to the bedrock down the scree slope (1–3 m). Additionally, they provided a means of VWC estimation, which agreed well with the values obtained using TDR and EC-5 sensors measured at IT1.

The soil characterisation provided in this study includes:

- Soil strength parameters derived from triaxial stress path test results,
- Additional curves of grain size distribution (GSDs) for soil extracted from IT1-4;
- A new set of measurements for in-situ unit weight, which are used later in the soil-specific calibration, and

- A contour line map with the depth bedrock in the scree slope, obtained through geophysical GPR techniques.

### 2.2.1. Soil Unit Weight and Grain Size Distribution (GSD)

The unit weight obtained from in situ measurements in this research used the balloon method device (Magdeburger Prüfgerätebau GmbH (HMP), DIN 18125-2:2011-03)) described in [6], (Supplementary Material, Figure S1). The dry unit weight was determined using the calculated volume and the dry weight of the soil excavated. There was one preliminary measurement, followed by two campaigns (November 2015) [6], and a second test/series performed in November 2016. The results are presented herein.

The GSD test was performed according to Swiss standard classification (SN 670004-2NA) for each trench location at IT1-4, using soil samples of weight between 30–38 kg (Figure 4). The soil was then used for the site-specific recalibration of the VWC.

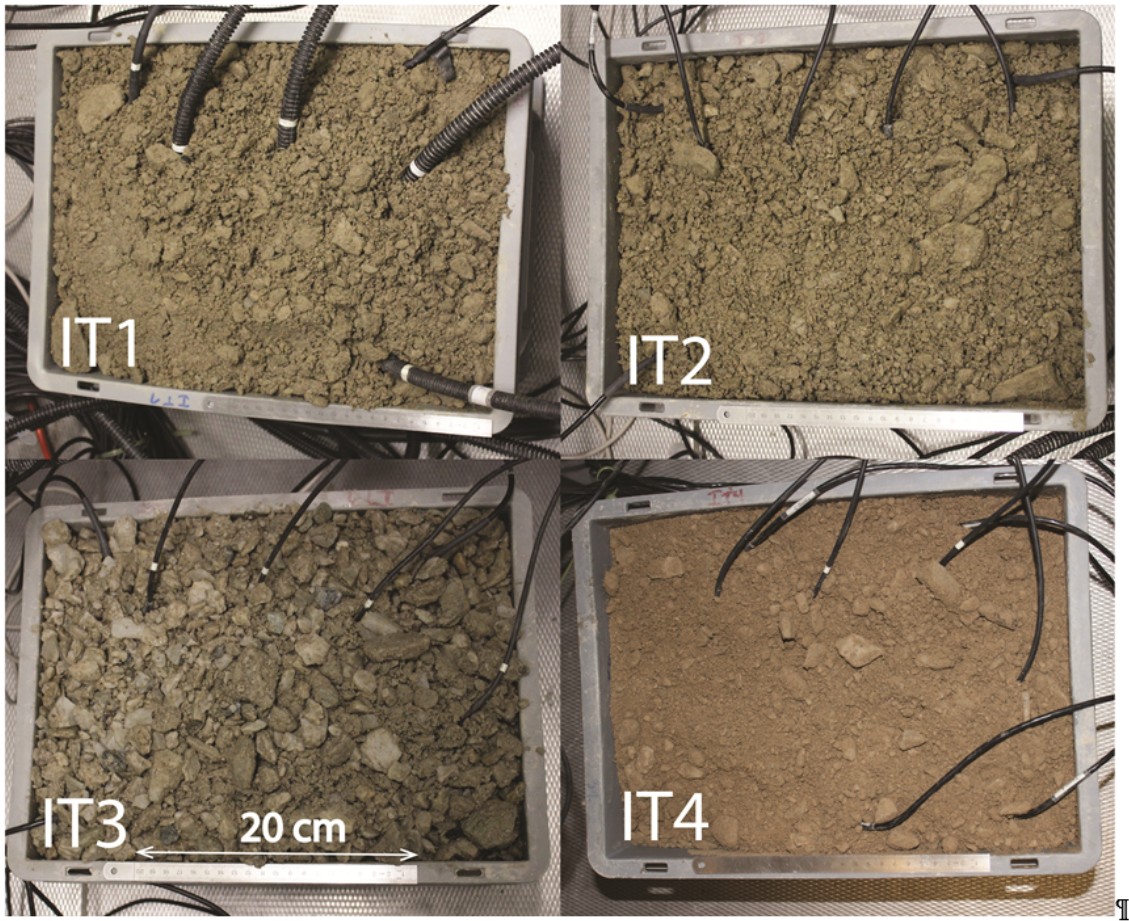

**Figure 4.** Recalibration of sensors: soil in reconstituted samples for instrumented trench (IT)1-4, with a moisture content is 8% at 1 °C. The boxes have a length and width of 36 and 26 cm, respectively.

### 2.2.2. Soil Shear Strength Parameters

Representation of Field Conditions

A slope stability analysis requires the study of the soil behaviour under the potential stress and stress path conditions that can lead to failure. Constant shear stress drained (CSD) triaxial tests (Figure 5a) on reconstituted specimens [32,33] reproduce the field conditions anticipated during infiltration and loss of effective stress [34] in terms of the saturation process in a steep slope, where

the angle of inclination is close to the angle of repose. This allows stress path dependent critical state parameters of the gravel to be determined under saturated conditions. The initial principal stress state ratios $K_c$ at point B in Figure 5a are based on the assumption of the principal stress orientation, as described in [35], for an infinite and planar slope. However, initial principal stresses in the triaxial devices could not exactly reproduce field conditions, since they are limited to being axial and radial. The pre-shearing anisotropic consolidation plus, a CSD stress path, could increase the $K_c$ until failure occurred (path B to C; Figure 5a).

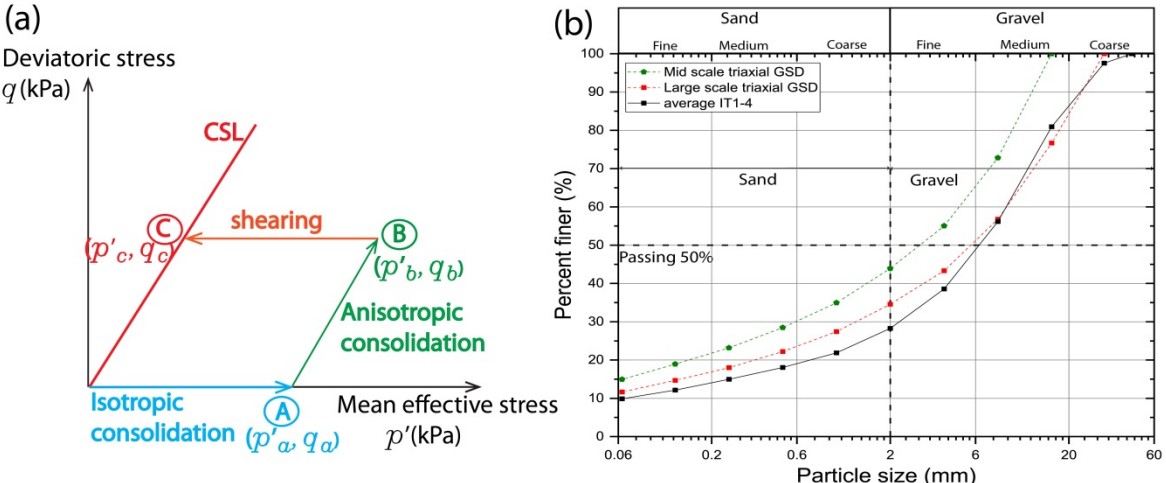

**Figure 5.** (**a**) Constant shear stress drained path (CSD). A is the point at the transition from isotropic to anisotropic consolidation, and B symbolises the transition from the latter to shearing as pore water pressures increases. Failure occurs at C, after shearing at constant deviatoric stress. (**b**) Grain size distribution (GSD) of the sandy gravel specimen: mid-scale (150 mm diameter) and large scale (250 mm diameter) and the average GSD of the IT1-4.

A series of CSD triaxial tests were carried out using a prototype gravel mixture with a representative GSD [36], which is an average of the soil from IT1-4 (Figure 5b). The soil response was compared and discussed for different device scale to particle diameter ratios, leading to determination of appropriate shear strength parameters of the reconstituted gravelly scree soil.

Triaxial Stress Path Testing

For a soil element below the water table, following a CSD path under saturated conditions, matches the process of water infiltration into overlying unsaturated layer. The pore water pressure rises at depth and hence the stress state ratio at constant deviator stress is increased until failure occurs [34].

The initial anisotropic stress state ratio $K_c = (\sigma_1')/(\sigma_3')$ in a steep scree slope indicates that it is in an incipient state of failure [34,37,38]. Pore water pressure (PWP) increases when water infiltrates into unsaturated soil, which occurs primarily under constant total stress, which reduces suction and hence the effective stresses. This can lead to failure under saturated or unsaturated conditions [34].

The analysis described herein has been applied in past slope stability studies e.g., [39,40], in helping to understand the failure mechanism. The constant shear stress drained path (CSD) is shown in Figure 5a, and follows three phases: isotropic consolidation (A), anisotropic consolidation (B), and constant shear stress path to failure (C). The CSD stress path can be achieved in IGT apparatuses in two different ways:

- Decreasing the mean effective stress $p' = (\sigma_1' + 2\sigma_3')/3$ at a constant rate, by reducing the total stresses, and keeping constant back-pressure and deviatoric shear stress $q = \sigma_1 - \sigma_3$ [41] for a 250 mm diameter specimen, after [34].
- Increasing back pressure, while total stresses are held constant.

The first method was used for the CSD in the large, and the second in the mid-scale, triaxial test. The loading piston can move independently in the large triaxial apparatus so that a decrease in cell pressure is possible, whereas the head plate is rigidly connected to the loading piston in the mid-scale apparatus, so it is not possible to apply cell pressure changes to affect the vertical total stress.

An alternative stress path to CSD path would be CADCAL, as applied by Casini [42] in a 50 mm diameter specimen of silty sand, where the consolidation phase is exclusively anisotropic, to an initial $K_c$ (B in Figure 5a), with subsequent shearing at constant q, under stress controlled conditions.

Soil Specimen Preparation

The two types of specimens were reconstituted for mid- and large-scale triaxial apparatuses (Supplementary Material, Table S1, Figure S2) with natural soil using an adopted representative GSD (Figure 5b) suitable for each test diameter. The maximum particle size for the specimens followed the CEN ISO/TS 17892-9:2004 for soils with a coefficient of uniformity ($C_u$) larger than 5, which limited the maximum grain size to 1/6 of the specimen's diameters. Therefore, the maximum particle sizes allowed for the mid- and large-scale triaxial specimens were 25 and 41 mm, respectively. A maximum size was of 16 and 31.5 mm was adopted respectively in both medium (150 mm diameter) and large (250 mm diameter) scale triaxial apparatuses on reconstituted specimens [36].

Fractions of each grain size, excluding those larger than 16 and 31.5 mm, were mixed carefully to make the mid- and large scale specimens, respectively (Figure 5b; Table 1). The specimens were prepared with initial moisture content of 3% at medium relative densities (27%–60%; Table 2), by moist tamping of 6 layers of 50 mm depth for the mid-scale and in 5 to 10 layers of 100 and 50 mm depth respectively for the large-scale.

**Table 1.** Soil characteristics for the reconstituted triaxial specimens.

| Test | Mid-Scale | Large Scale |
|---|---|---|
| $D_{10}$ (mm) | 0.04 | 0.04 |
| $D_{30}$ (mm) | 0.60 | 1.31 |
| $D_{60}$ (mm) | 5.01 | 9.10 |
| $D_{max}$ (mm) | 16.00 | 31.50 |
| $C_c$ (-) | 1.90 | 4.70 |
| $C_u$ (-) | 131.90 | 227.50 |
| Percentage of fines (%) | 14.9 | 11.7 |
| Swiss Standard Classification (SN 670 004-2b NA) | GM | GM |
| Ratio $D_{triaxial}/D_{max}$ | 150/16 = 9.38 | 250/31.5 = 7.94 |
| $e_{min}$ (-) | 0.269 | 0.341 [1] |
| $e_{max}$ (-) | 0.570 | 0.691 [1] |

[1] $e_{max, min}$ from IT1 field soil.

**Table 2.** Triaxial testing programme on gravelly scree soil: specimen preparation.

| Test | $e_0$ (-) | $D_{r,i}$ (%) | $K_c = \sigma'_{1,B}/\sigma'_{3,B}$ (-) | $q_A$ (kPa) | $p'_A$ (kPa) | $\sigma'_{1,B}$ (kPa) | $\sigma'_{3,B}$ (kPa) | $q_B$ (kPa) | $p'_B$ (kPa) |
|---|---|---|---|---|---|---|---|---|---|
| mid_1 | 0.49 | 27 | 1.83 | 6.0 | 40.0 | 143.1 | 78.0 | 65.1 | 99.7 |
| mid_2 | 0.52 | 17 | 1.94 | 6.0 | 19.0 | 72.6 | 37.4 | 35.2 | 49.1 |
| mid_3 | 0.54 | 10 | 1.88 | 5.0 | 30.0 | 108.3 | 57.5 | 50.8 | 74.4 |
| large_1 | 0.53 | 46 | 1.74 | 8.0 | 36.0 | 133.4 | 76.5 | 57.0 | 95.5 |
| large_2 | 0.48 | 60 | 1.85 | 5.0 | 18.0 | 65.4 | 35.3 | 30.1 | 45.3 |
| large_3 | 0.59 | 29 | 2.42 | 3.3 | 59.0 | 227.4 | 94.1 | 133.3 | 138.6 |

$e_0$: initial void ratio; $D_{r,i}$: relative density; $K_c$: principal stress ratio.

Testing Programme

The testing programme is presented in Table 2 and includes the characteristics of the stress paths for the mid-scale triaxial (mid) and the large-scale triaxial (large) devices. The initial conditions before consolidation are given by the void ratio ($e_0$) and relative density ($D_{r,i}$). The stresses are defined after each consolidation phase by $p'_A$ and $q_A$ for isotropic consolidation and by $p'_B$ and $q_B$ for anisotropic consolidation (Figure 5a). The calculated stress state ratio ($K_c$) captures the anisotropic stress state before the shearing phase starts.

Theoretical Framework: Dilatancy

According to Bolton [43], the dilatancy angle ($\psi$) in granular soils is represented by $\phi'_{max} - \phi'$, where $\phi'_{max}$ is the peak friction angle, which can be calculated for triaxial conditions as:

$$\psi = 3 \times I_R, \text{ with: } \psi \ (°) = \text{dilatancy angle; } I_R = \text{dilatancy index} \tag{1}$$

where as the dilatancy index is:

$$I_R = I_D \ (Q - ln(p')) - 1, \text{ with: } I_R = \text{dilatancy index;}$$

$$Q = \text{natural logarithm of the mean stress } p_c \text{ at which dense soil first reaches the normal consolidation line} \tag{2}$$

$$I_D \ (\text{-}) = \text{relative density}$$

According to Bolton [44], $Q$ can be estimated to be 9.2. The relative density is determined with the void ratio as:

$$I_D = \frac{e_{max} - e}{e_{max} - e_{min}}, \tag{3}$$

with: $e_{max}$ (-) = maximum void ratio, $e_{min}$ (-) = minimum void ratio, $e$ (-) = void ratio.

2.2.3. Ground Model: Ground Penetrating Radar (GPR)

Methodology, Data Acquisition, and Data Processing

Ground penetrating radar (GPR) was carried out to characterise the scree slope (Figure 1b) in the Bochtür area of the Meretschibach catchment. This geophysical method was used to complement the long-term geotechnical seasonal field monitoring at this site.

GPR uses high frequency electromagnetic waves to map anomalous electrical structures in the subsurface. Transmitting and receiving antennas are kept at a fixed distance and are carried along the surface. Radar pulses are emitted into the ground by the transmitter, partly reflected and transmitted at discontinuities where electromagnetic properties (primarily the dielectric permittivity) of the subsurface material change (as is usually the case at the soil-bedrock interface). The reflected signal is captured by the receiver antenna. The time required for the signal to travel from the surface to the bedrock and back to the surface (two-way travel time) and can be converted to depth, using the propagation velocity of the GPR waves. More information on the GPR theory, resolution and penetration can be found in [45–49].

The choice of the antenna frequency is crucial to determine the penetration depth and resolution, which are always a trade-off: the higher the frequency, the higher the (vertical and horizontal) resolution, but this comes at the expense of a decreased penetration depths [45,47–49]. The 250 MHz antenna from the PulseEKKO manufacturer was chosen for the main GPR acquisitions in this project, allowing for a vertical resolution of up to 0.1 m and a depth penetration of up to 10 m. The relevant acquisition parameters used during GPR surveys are provided in Supplementary Material Tables S3 and S4 [31].

GPR acquisition was carried out on 15 and 16 May 2014. The steepness of the terrain made walking difficult and the accessibility of the active channel proved to be too dangerous on foot. Therefore, 20

profiles (Figure 1b) were recorded on the scree slope using a large number of stacks (Table S3), because the slow walking ensured enough traces were recorded in each position.

A relatively standard processing workflow was applied to the GPR data acquired. It included (i) band-pass filtering to enhance the signal-to-noise ratio, (ii) removal of system ringing with singular value decomposition filter (e.g., [50]), (iii) trace binning for obtaining equidistant traces (0.05 m), (iv) FX-deconvolution to enhance lateral continuity and (v) time-to-depth conversion using a velocity of 0.1 m/ns.

### 2.3. Seasonal Field Monitoring

The seasonal alpine climate conditions directly affect the VWC and temperature measured in the soil. The seasonal response exhibited the following trends over the years of field monitoring:

- Soil temperature changes are more noticeable at shallow depths, with a higher diurnal variation in summer;
- As temperature drops in autumn, a winter regime develops with temperatures around 0 °C (or lower), with VWC reaching minimum annual values (0–0.07), and a persistent snow layer in place over several months, which insulates the underlying gravel from temperature and VWC variations;
- As temperature increases in spring, leading to snow-melt, the ground resaturates and the VWC rises;
- Subsequent VWC changes in the summer season are directly related to rainfall events, as well as the GSD of the trench location, and the unsaturated hydraulic conductivity of the soil.

Final results from the seasonal monitoring fieldwork are presented, now including the complete records of three years of VWC, temperature and precipitation, including a soil-specific site calibration for the estimation of the VWC under variations in temperature. A selection of data is used subsequently in the numerical preliminary simulation of slope instabilities using SLOPE-SEEP/ W [51] to evaluate the hazard.

### 2.3.1. Site-Specific Calibration

The first two years of data were presented in [6], where some differences were observed in VWC measurements between capacitance and TDR sensors at similar depths and locations, were discussed but not yet investigated. The dielectric properties of the soil (related to the VWC) are strongly affected by the water content, soil structure and density [52], calibration of the sensor [53], input voltage [54], orientation and volume of measurement (depending of the type of sensor), and temperature [52,55].

It was necessary to investigate whether there was any influence from fluctuations in daily temperatures, especially for surficial soil measurements, and for freezing and thawing [6]. The effect of temperature on the VWC has been already mentioned in the literature by several authors. On the one hand, Bogena [54], used an EC-5 in a known liquid permittivity at a range of temperatures (5 °C–40 °C) and found that the sensor generally showed an increase in VWC, with an increase in temperature. On the other hand, Topp [52] measured the apparent dielectric constant (related to VWC) using a TDR in clay loam in a temperature range of (10 °C–36 °C) and stated that there was no significant temperature dependence. Pepin [56] used TDRs to study the effect of temperature in VWC in sand and peat and found that the dielectric constant decreased with increases in temperature, and suggested the use of empirical or theoretical relationship corrections due to temperature effects on the composite dielectric constant for higher VWCs. As no defined trend in the effect of temperature in the soil has been found and no one has referred to a gravelly soil, a site-specific calibration with maximum sizes passing the 45 mm sieve was carried out for the seasonal variation in temperature, which is presented in the Supplementary Material, Figures S4–S14.

### 2.3.2. Monitoring Instruments

Each instrumented trench contained time domain reflectometry (TDR100; Campbell Scientific) and capacitance sensors (EC-5/10HS; Decagon Devices) to determine the VWC by means of the measurement of the apparent dielectric constant /permittivity. The VWC was complemented by temperature and precipitation data from two meteostations. Detailed information about the sensors and meteostations can be found in [6].

A site-specific calibration was performed (Figure 4) to verify whether the VWC values at similar depths and location will improve in terms of accuracy for each IT (1–4), when compared with the original calibration, which was done at room temperature with the soil from IT1 only. Soil sampled from the field at each of the IT locations was used for this recalibration, over a temperature range variation (−5 °C to room ambient temperature).

Each sensor was calibrated in the laboratory after it had been removed from the field, under a programme of controlled moisture and temperature conditions, simulating the changes in weather conditions for every season. Although some of the sensors were broken after three years in the field, and some of them failed during the recalibration process back in the laboratory, it was possible to record information from the remainder (TDR, EC-5, 10HS) to analyse the effect of temperature changes in the sensor performance, when these were installed in gravelly soil at similar relative densities.

### 2.3.3. Hydro-Mechanical Effects of Vegetation on Slope Stability

Vegetation in slopes contributes to stability in two ways. Firstly, the contribution of the roots to reinforcing the soil matrix is most valuable [57,58] until they pull out or break. It is also useful to understand changes of volumetric water content due to evapotranspiration during summer months or removal of above-ground vegetation, as well as their influence on the soil behaviour. Two examples illustrate these effects.

Smethurst et al. [59] developed a 6 years field monitoring experiment for a 7 m high, vegetated railway embankment on the Shenfield-Southend line in Essex, UK. It had been constructed in 1887 and consisted of over a metre of ballast overlaying ash, clay fill and the natural London clay. The vegetation enhanced the shrinkage and swelling of the soil, so the authors studied the influence of the tree removal and its effect on the soil stability. The tree roots could reach depths of up to 2–3 m, whereas grass roots penetrated up to 1 m depth. Therefore, grass would uptake from surficial depths, where the soil is more exposed to rain infiltration and weather changes, whereas trees roots could reach the foundation of the embankment, creating persistent suction zones. Many native deciduous British species uptake water during summer, drying the soil when transpiration is more than the rainfall and ceasing any transpiration during winter time, which also reduces the soil suction. The VWC and pore water measurements during transitional season periods (March, September) were measured and compared before and after tree removal. The result showed that the VWC content varied seasonally at shallower depths (up to 2.25 m) when trees were present, and remained dryer at greater depths. Removal of the trees reduced the evapotranspiration and increased infiltration at shallower depths with an increase the VWC at greater depths. In the case of PWP, trees induced higher suctions at all depths, while tree removal resulted in an increase of PWP with saturation. The presence of vegetation increased suction and therefore the stability of the soil.

Yildiz et al. [60] investigated the shearing behaviour of root-permeated moraine, which exhibited dilatancy under saturated conditions. The moraine from a subalpine landslide in Dallenwill, canton Nidwalden, Switzerland was sieved to maximum size particle of 20 mm. The samples (500 × 500 × 400 mm) containing 96 plants of two different categories with 6 months of growth, were tested in an inclinable large-scale direct shear apparatus at different normal force loads, simulating depths of shallow landslides. After shearing, the roots were weighed and soil strength parameters obtained. Vegetated soils, which exhibit dilatancy, are expected to mobilise higher peak shear strength due particle interlocking and the tensile strength of their roots. A decrease in shear strength after a peak, then can be due to root breakage or pulling out, and/or due to particle breakage or abrasion

after dilatancy. The results of the study showed when the soil reaches peak strength with a maximum dilatancy angle, it had not necessarily mobilised the maximum tensile strength in the roots, but a combined approach of taking dilatancy and root reinforcement into consideration results in more realistic quantification of the soil strength behaviour. He also quantified the influence of suction on the strength.

Stunted vegetation in the Meretschibach-Böchtur catchment, such as spruce (Picea abies) and low shrubs of less than 1 m height, are only present partially along the slope, due to the active erosion and mass movements that impact on them and destroy them, increasing the risk of remobilisation of the debris. Some of the past mass movements have been summarised by Oggier [5] and Fankhauser [31].

The integration of vegetation root reinforcement, plus the potential dilatancy of the gravel in a site location with extreme alpine climate conditions increase the complexity of the problem. As this project focused on the ground characterisation and the mechanism of failure, the effects of the root reinforcement is not quantified in this study and therefore its considered to be a conservative approach when testing the fallow soil shear strength.

## 2.4. Preliminary Numerical Modelling

A landslide has not been observed during three years of seasonal field monitoring in the scree slope. The field measurements, including the geophysical monitoring of the site, contributed to the characterisation of the bedrock geometry. Moreover, the measurements of the volumetric water content, as well as the pore water pressure provided an image of the hydraulic responses of the scree slope to rainfall events. These measurements also contributed to an understanding of the groundwater flow, and the unsaturated state of the soil.

The degree of saturation in an alpine scree slope is influenced by the seasonal weather variations [6]. In warmer seasons, the snow-melt and rain infiltrates into the ground and increase the soil saturation, therefore, it was hypothesised that the mechanism of failure would be due to saturation of the soil and subsequent reduction of suction and effective stress. This was used as a basis for this preliminary analysis of the failure mechanisms.

The field situation studied during three years of seasonal field monitoring, gave indications about the possible mechanisms of failure. Simplification of this complex interaction was developed in the ground model. The soil characterisation reported in Lucas et al. [6], the laboratory testing including the strength parameters from stress path triaxial testing, and the field monitoring datasets were the input to the numerical model of this study.

Preliminary numerical simulations into the slope stability were performed using the program SEEP-SLOPE/W [51] on a simplified prototype of the scree slope, consisting of a representative gravel layer and varied bedrock geometries which were then subjected to groundwater flow. The objective was to determine which bedrock geometry represents the most hazardous scenario under similar groundwater hydraulic conditions.

### 2.4.1. Model

The model (Figure 6) consisted of a unique layer of homogeneous gravel of 2.5 m thickness underlain by impermeable bedrock that could be either parallel to the slope or with a bedrock step. The gravel parameters were derived from the soil characterisation in this study from Tables 1–6. Four different types of slope-bedrock geometry were tested: (a) bedrock parallel to slope with toe, (b) bedrock parallel to slope with no toe, (c) bedrock with a single step and toe and, (d) bedrock with a single step and no toe. Although the scree slope length is of the order of the 600 m (Figure 1b), the model was simplified to 50 m for this preliminary analysis, and the angle of inclination was selected as 40°, within the range of the slope inclination reported for the scree slope [6].

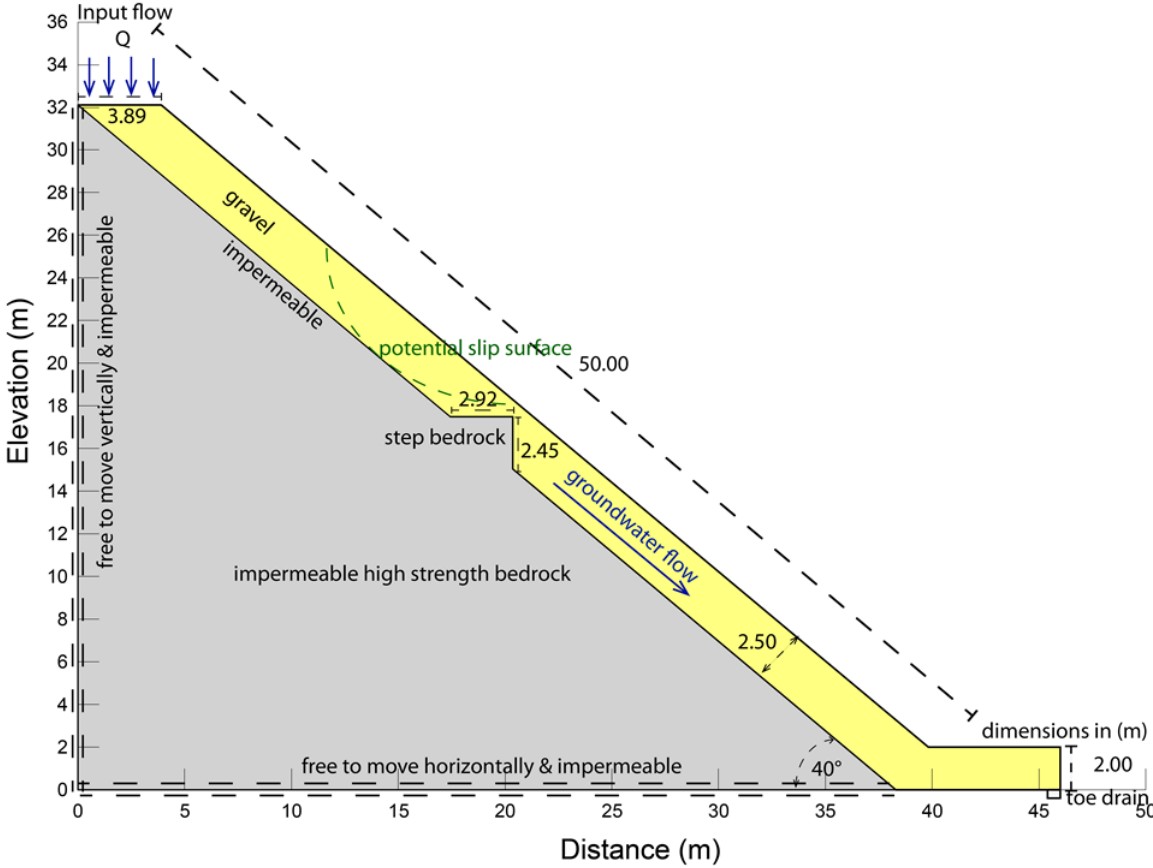

**Figure 6.** Model slope-bedrock geometry for the numerical modelling with bedrock step and toe (type c) with units in m.

**Table 3.** Soil properties for the numerical simulations with SEEP/W and SLOPE/W.

| Parameter | Value |
|---|---|
| Soil Classification | GP-GM |
| Internal Friction Angle $\phi'$ | 41° |
| Specific Density (kg/m³) | 2.68 |
| Cohesion $c'$ (kPa) | 1 |
| Hydraulic Conductivity $k$ (m/s) | $1.30 \times 10^{-5}$ |
| Dry Unit Weight $\gamma$ (kN/m³) | 19.42 |
| Void Ratio $e$ (-) | 0.38 |

$k$: hydraulic conductivity from the Kozeny—Carman equation [61] average from field data IT1-4 [6]; $\phi'$:derived from triaxial stress path test; $\gamma$, $e$: derived from in situ measurements.

**Table 4.** Dry unit weight calculated from in-situ testing.

| Trench | Final Campaign November 2016 | | Dry Unit Weight Used in Site-Specific Recalibration | | |
|---|---|---|---|---|---|
| | Void Ratio (-) | Dry Unit Weight (kN/m³) | Void Ratio (-) | Dry Unit Weight (kN/m³) | $D_{max}$ (mm) |
| IT1 | 0.29 | 19.95 | 0.32 | 19.91 | 31.5 |
| IT2 | 0.38 | 18.75 | 0.40 | 18.84 | 60.0 |
| IT3 | 0.38 | 18.73 | 0.41 | 18.61 | 31.5 |
| IT4 | 0.32 | 19.54 | 0.36 | 19.32 | 31.5 |

**Table 5.** Soil classification of scree soil from the instrumented trenches IT1-IT4.

| Trench | $D_{10}$ (mm) | $D_{30}$ (mm) | $D_{60}$ (mm) | $C_c$ (-) | $C_u$ (-) | Percentage of Fines (%) | Swiss Standard Classification (SN 670 004-2b NA) |
|---|---|---|---|---|---|---|---|
| IT1 | <0.06 | 1.525 | 7.70 | - | - | 11.50 | GP-GM |
| IT2 | 0.075 | 2.075 | 8.35 | 6.87 | 111.3 | 9.43 | GP-GM |
| IT3 | 1.010 | 6.05 | 11.45 | 3.16 | 11.3 | 4.97 | GP |
| IT4 | <0.06 | 1.1 | 6.40 | - | - | 13.61 | GM |

**Table 6.** Summary of constant shear stress drained (CSD) triaxial stress path test results at failure from an adopted grain size distribution (GSD) of the gravelly scree.

| Test | $D_{r,i}$ (%) | $\sigma'_{1,C}$ (kPa) | $\sigma'_{3,C}$ (kPa) | $q_C$ (kPa) | $p'_C$ (kPa) | $\varepsilon_{a,C}$ (%) | $\varepsilon_{v,C}$ (%) | $M$ (-) | $\phi'$ (°) |
|---|---|---|---|---|---|---|---|---|---|
| mid_1 | 27 | 77.0 | 16.3 | 60.8 | 36.5 | 4.35 | −0.66 | 1.66 | 41 |
| mid_2 | 17 | 40.3 | 7.8 | 32.5 | 18.7 | 3.43 | −0.53 | 1.74 | 42 |
| mid_3 | 10 | 60.5 | 13.1 | 47.4 | 28.9 | 3.07 | −0.35 | 1.71 | 42 |
| large_1 | 46 | 61.2 | 8.7 | 52.5 | 26.2 | 4.21 | −0.80 | 2.00 | 49 * |
| large_2 | 60 | 32.8 | 2.1 | 29.7 | 12.0 | 1.41 | −0.54 | 2.50 | 61 * |
| large_3 | 29 | 156.5 | 32.0 | 124.5 | 73.5 | 6.32 | −0.11 | 1.70 | 41 |

\* These values include dilatancy.

### 2.4.2. Hydraulic Conditions

Input flow ($Q$) to trigger instability was applied to the top of the slope for each geometry case. This groundwater flow antecedent increased the saturation in the well-drained gravel in the steep slope and hence the PWP decreasing the effective stress of the soil. Although the variation of PWP due to water infiltration is a time dependent process, a simplified steady state analysis was conducted using SEEP/W with saturated/unsaturated analysis to impose the hydraulic condition for the further slope stability analysis.

### 2.4.3. Slope Stability Analysis

Slope stability simulations were performed in 2D using the SLOPE/W program based on the Morgenstern-Price method [51] and the hydraulic conditions imposed by SEEP/W. A slip surface failure is determined for a factor of safety (F.S.) = 1. Due to the increase in PWP, the effective confining stress in the slope decreases, triggering the failure. The results were summarised in a figure for different slope geometries with the flow rate required to cause an instability in a gravelly soil, and corresponded to a preliminary and simplified analysis.

## 3. Results

### 3.1. Characterisation

#### 3.1.1. Dry Unit Weight

Two campaigns of in situ measurements were performed to determine dry unit weight, the first presented in [6], and the second performed in 2016 (Table 4). It was extremely challenging to carry out such tests in the field because of the steep slopes (mostly between 33°–43°), which contained coarse gravel and cobbles, and which in many cases led to unstable walls when excavating the nominally cylindrical void. This volume often exhibited an irregular shape due to stones crossing the boundary of the circular ring around the excavation; therefore, the volume excavated was often less than that defined in the test specification.

Notwithstanding these challenges, sixteen densitometer tests were completed in total during the seasonal field monitoring programme. One single preliminary test was carried out prior to November 2015, a few metres downhill of IT4 in a relatively flat area, leading to a unit weight of 19.11 kN/m$^3$. A second test in November 2015 was discarded, because of failure of the excavation during testing. Later, a campaign of 9 tests was performed in July 2016 [6] in the steep slope, which affected the volume of measurement due to the reasons mentioned previously. A final campaign of 5 tests was achieved in November 2016, after searching for suitable locations where the densitometer tests could be carried out closer to the norm, in terms of uniform volume cylinder. The results are presented in Table 4.

### 3.1.2. Grain Size Distribution (GSD)

The GSD of the soil extracted from each instrumented trench (IT1-4) was determined and used for the 4 sets of site-specific VWC sensor recalibrations, with the same GSD profile and a $D_{max}$ = 31.5 mm and 60 mm for IT1, IT3-4 and IT2 respectively. The results and the GSD curves are presented in Figure 7 and the classification in Table 5, showing a mixture between poorly graded gravelly scree with sand and silt, with more fines in IT1 and IT4.

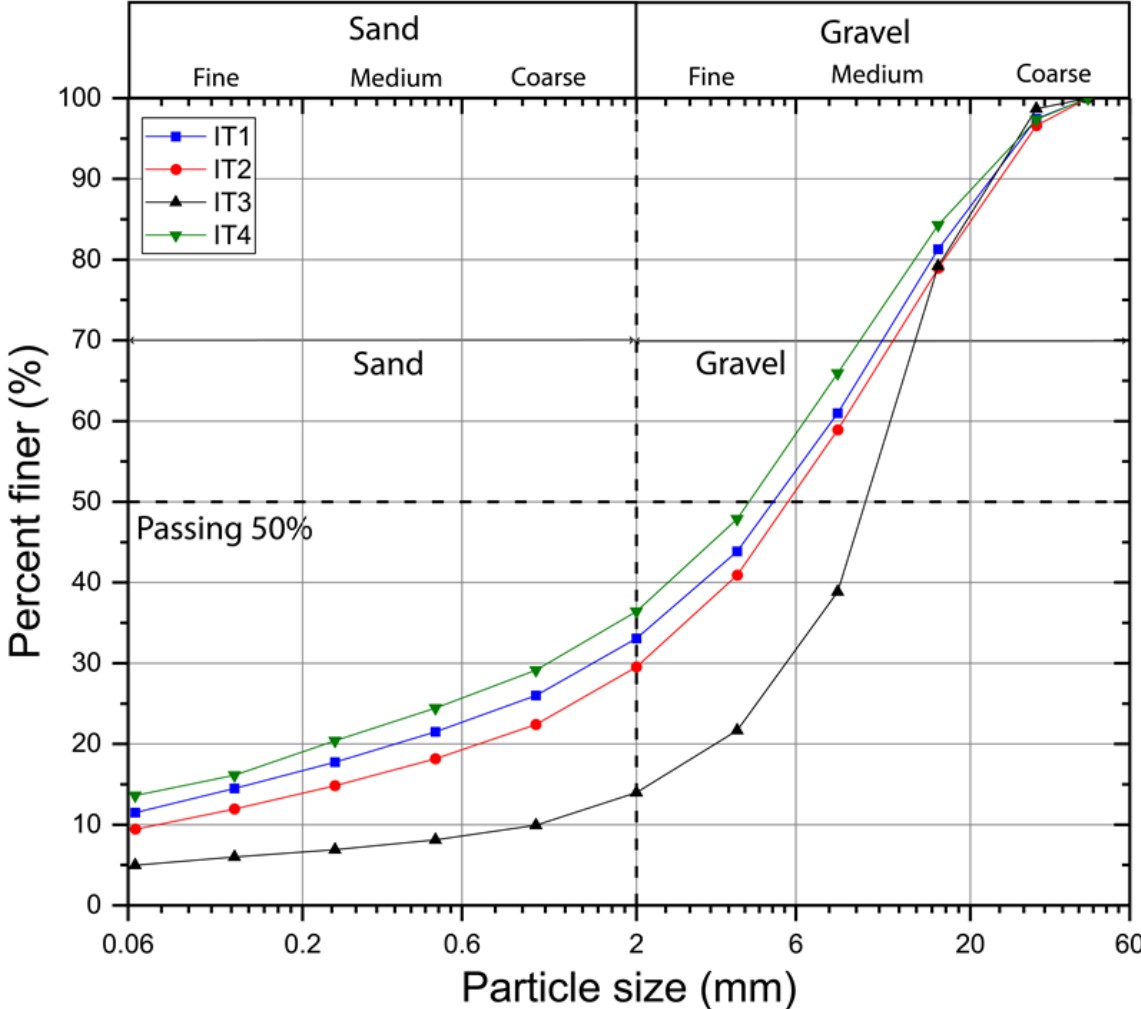

**Figure 7.** Grain size distribution (GSD) site-specific calibration soil instrumented trenches IT1-IT4 after site-specific calibration process.

### 3.1.3. Triaxial Stress Path Testing: CSD Results

Results of the CSD tests are shown in Figure 8, in terms of $\varepsilon_a$ versus $p'$, $\varepsilon_v$ versus $p'$ and $q$ versus $p'$. Table 6, shows the stresses $p'_C$ and $q_C$ at failure, which were used together with the slope in $p' - q$ space, $M = (6 \times \sin\phi')/(3 - \sin\phi')$, to obtain a friction angle at failure $\phi'$ of 42° and 41°, for the mid- and large-scale specimens respectively, with zero cohesion ($c' = 0$) in both cases.

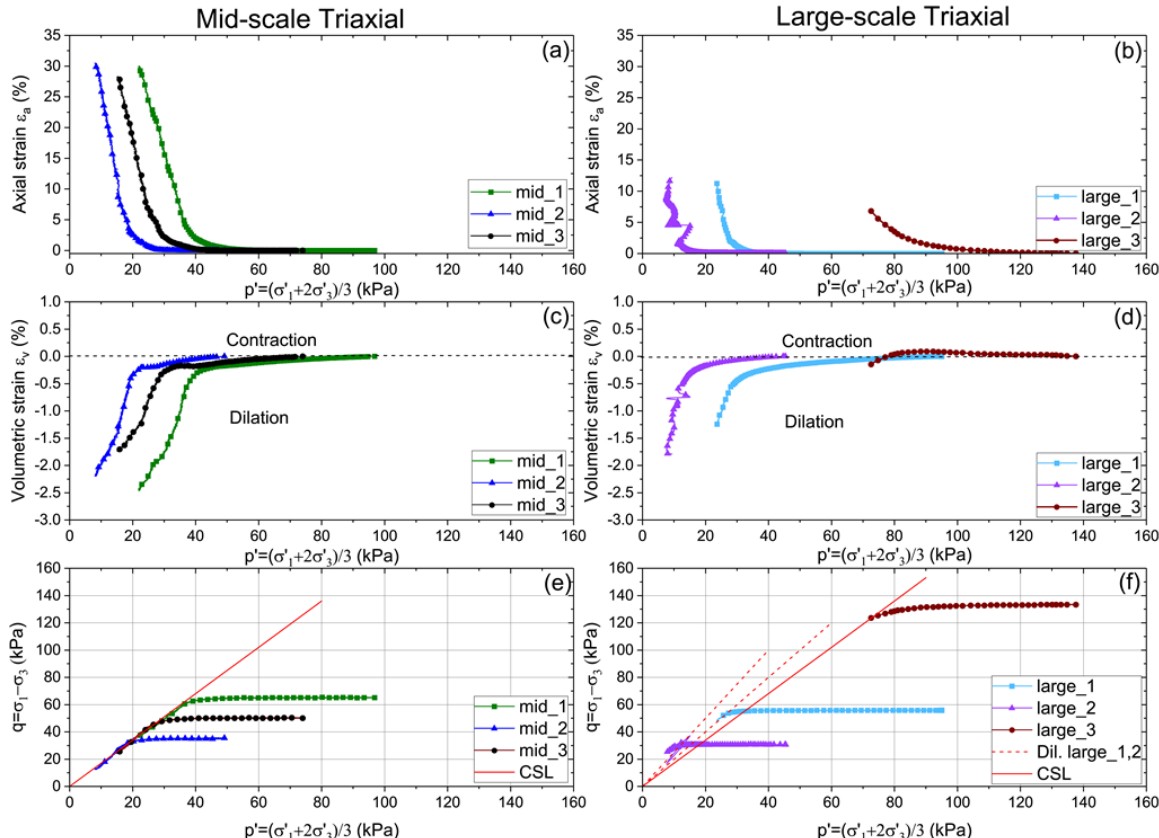

**Figure 8.** Constant shear stress drained test (CSD) results during shearing on gravelly scree soil (medium and large, with maximum particle size of 16 and 31.5 mm respectively). (**a**,**b**) axial strain versus mean effective stress; (**c**,**d**) volumetric strain versus mean effective stress; (**e**,**f**) mean effective versus deviator stress at constant deviator stress and showing the stress path and critical state line (CSL). The mid-scale triaxial test is on the left, the large-scale on the right.

Figure 8a,d illustrate steep increases of dilation and axial strain after the yield point of the gravelly soil, followed by initiation of failure. The soil behaviour is dilative in most cases (Figure 8c,d) with the exception of the test large_3, ($D_{r,i}$ = 29%) (Figure 8d), which contracts initially, and then dilates towards failure.

The stress paths shown in the $p' - q$ diagrams in Figure 8e,f, exhibit a horizontal constant deviator stress path until reaching failure, where the soil specimens no longer represent right cylinders and the calculated stresses decrease. A critical state line (CSL) was drawn through the origin and the stresses at failure in the mid- and large-scale tests Figure 8e,f. The mid-scale triaxial tests lead to derivation of a single CSL line, probably due to the finer soil grading (Table 1), which dominated the mobilisation of the soil strength. The large-scaled triaxial tests show different dashed lines (Figure 8f) of data representing the higher friction angles mobilised at low confining pressures, which are attributed to dilatancy (e.g., [43]) and the influence of the hardness of the coarser gravel on the shear strength. In this case, the values reported for the tests large_1 and large_2 do not represent the critical state conditions [43].

The dilatancy angle ($\psi$) (Equations (1)–(3)) was calculated for each of the triaxial tests (Table 7) in terms of the dilatancy index ($I_R$), mean effective stress ($p'$) and relative density ($I_D$), according to Bolton's [43] Equation (1). The dilatancy mobilised at nominal failure was found consistently to be larger at low confining pressures (Table 7). Theory suggests that the lower the confining stresses, the higher the relative density and grain crushing strength are, the greater the $I_R$ and hence $\psi$ will be [43]. The small angle of dilation obtained in test large_3, carried out at higher confining pressure, indicates the possibility of particle abrasion or crushing during testing [43,62–65]. This assumption was confirmed by comparing the grain size distribution before and after shearing, where an increase of approximately 4% in the fines content was observed in the post-test sieving due to shearing (Supplementary Material, Table S2, Figure S3).

**Table 7.** Calculated dilatancy in triaxial tests.

| Test | $e_0$ (-) | $I_{D,0}$ (-) | $e_B$ (-) | $e_C$ (-) | $I_{D,C}$ (-) | $p'_C$ (kPa) | $I_R$ (-) | $\psi$ (°) |
|---|---|---|---|---|---|---|---|---|
| mid_1 | 0.49 | 0.27 | 0.275 | 0.283 | 0.95 | 36.5 | 4.34 | 13.03 |
| mid_2 | 0.52 | 0.17 | 0.323 | 0.330 | 0.80 | 18.7 | 4.00 | 12.00 |
| mid_3 | 0.54 | 0.10 | 0.371 | 0.376 | 0.64 | 28.9 | 2.76 | 8.28 |
| large_1 | 0.53 | 0.46 | 0.364 | 0.375 | 0.90 | 26.2 | 4.36 | 13.07 |
| large_2 | 0.48 | 0.60 | 0.332 | 0.339 | 1.00 | 12.0 | 5.75 | 17.26 |
| large_3 | 0.59 | 0.29 | 0.581 | 0.593 | 0.28 | 73.5 | 0.37 | 1.12 |

### 3.1.4. GPR Results

Figure 9a,b shows an example of a processed GPR section. It is roughly parallel to the slope inclination (Figure 1b). Reflections from the bedrock interface can be identified quite clearly (blue arrows in Figure 9b, although they cannot be traced continuously along the entire section. All reflected signals have been picked from all profiles and the corresponding bedrock depths have been interpolated across the entire scree slope covered by the GPR profiles.

The resulting contour line map is shown in Figure 10. Each colour represents an estimated depth of the bedrock, varying from 0.2 m (blue) to 2.8 m (red). The contour map shows that IT1-4 are located, where the bedrock depth is between 0.6 to 1.6 m, and that the deeper location of the bedrock is oriented to the mid-west, towards the uphill section of the scree slope. This result suggests that the maximum volume that could be mobilised in a potential landslide on this slope (of triangular area with average depth of 1.5 m) would be approximately $0.5 \times 100 \times 200 \times 1.5 = 15{,}000$ m$^3$.

### 3.2. Seasonal Field Monitoring

### 3.2.1. Overview

The VWC and temperature were measured at specific locations of the slope (IT1-4). These data were complemented by the precipitation data from two meteostations. Figure 11 shows the data overview for VWC and temperature versus date measured in each trench (IT1-4) over a period of three years of seasonal field monitoring from November 2013 to October 2016. Each colour trace represents a VWC measured with a type of sensor (EC-5, 10HS and TDR), which has been recalibrated following reassessment of the relevant parameters. The depth was recorded in the sub-index for both VWC and temperature sensors. The plots are arranged in chronological order of installation and the precipitation recorded by two meteostations, IGT and WSL, is added at the bottom. Background colours of light blue and yellow differentiate between winter and summer regimes, respectively.

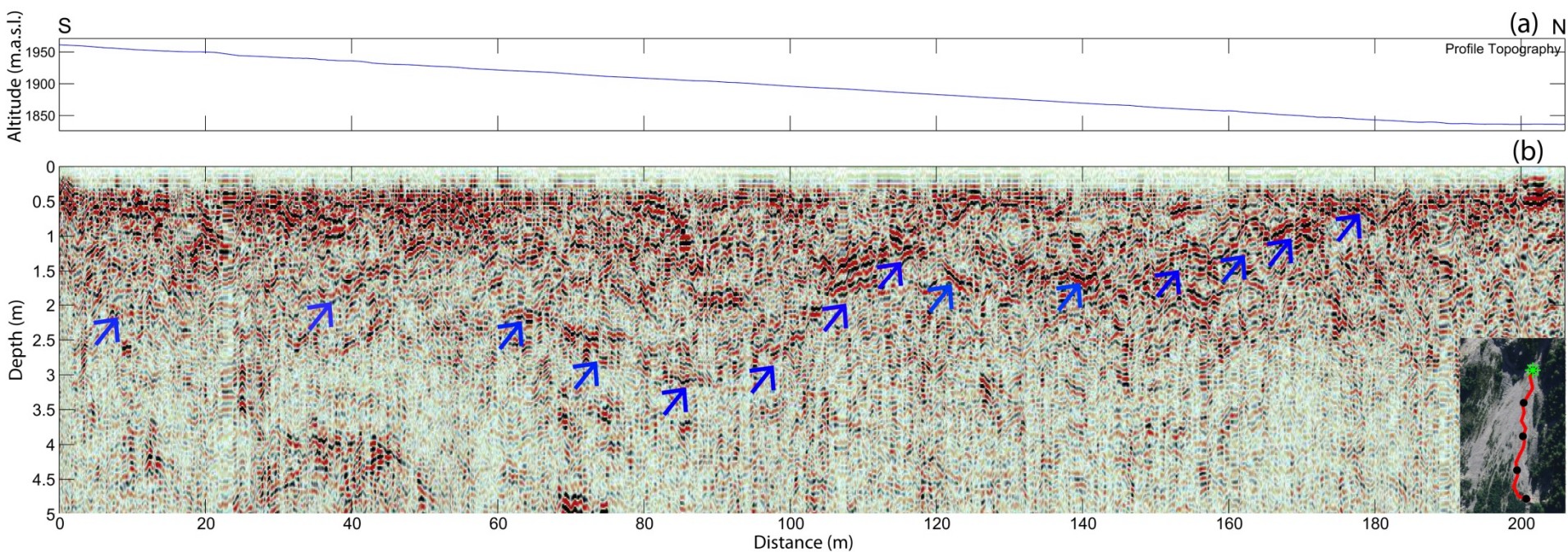

**Figure 9.** Resulting Ground Penetrating Radar (GPR) section from profile 8 (Figure 1b). (**a**) Ground elevation in longitudinal view; (**b**) Results after all processing steps were applied. Blue arrows indicate reflections from the bedrock interface.

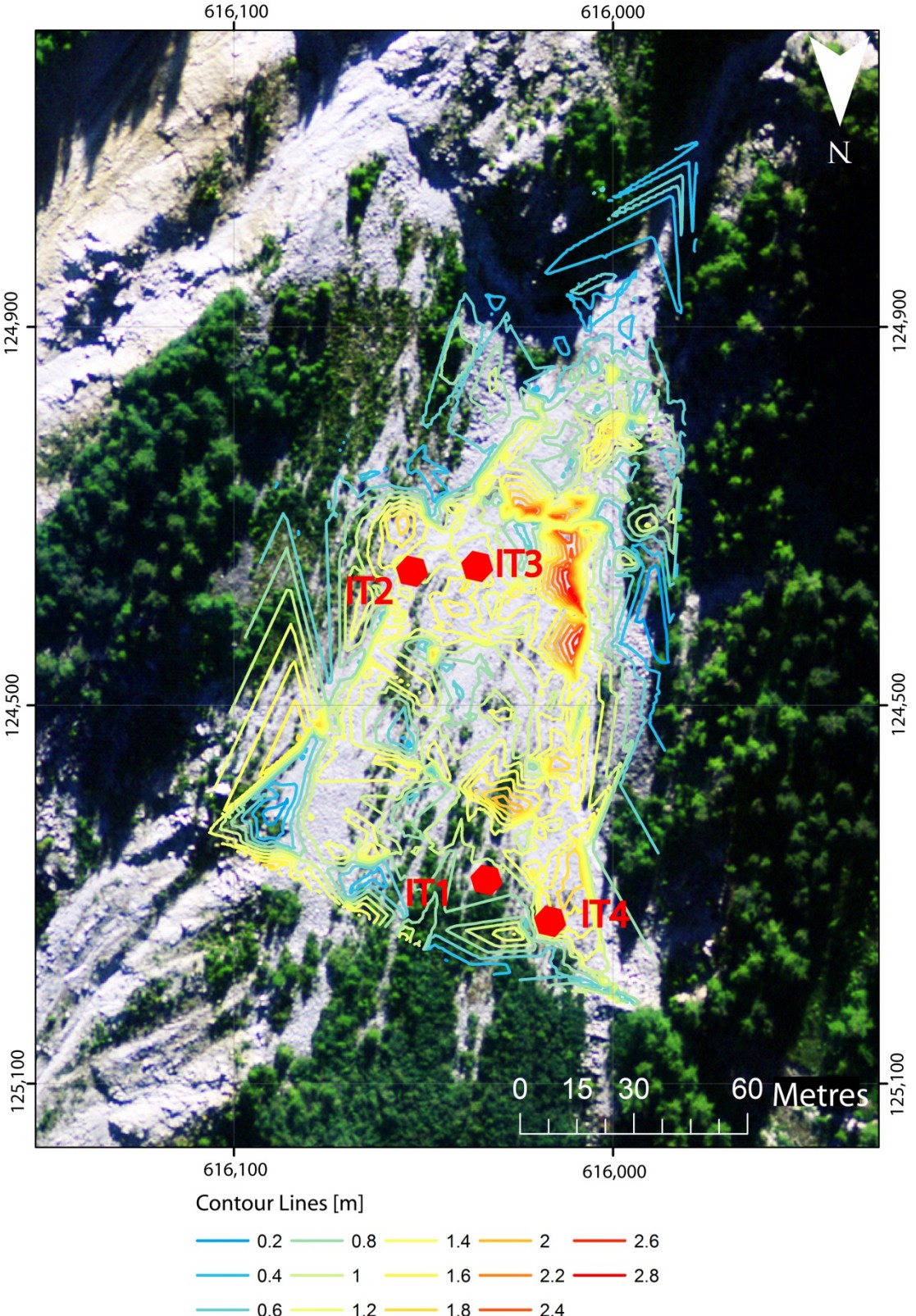

**Figure 10.** Contour lines of depth to bedrock in metres.

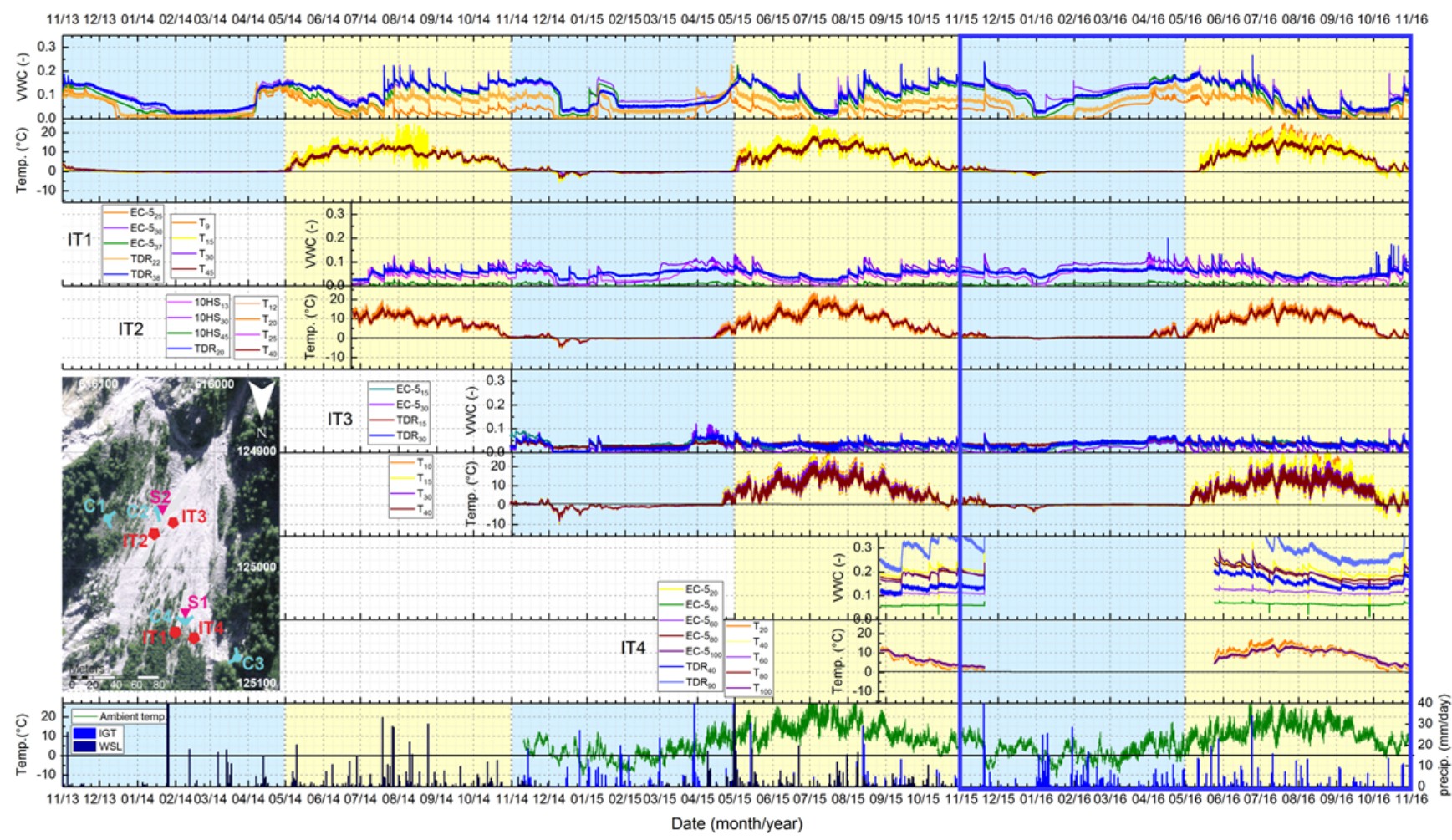

**Figure 11.** Volumetric Water Content (VWC) & Temperature IT1-4. Overview of measurements at instrumented trenches IT1-4 in the Meretschibach catchment for three years duration of seasonal field monitoring (November 2013–October 2016): VWC, temperature with precipitation data at the bottom.

### 3.2.2. New data: November 2015–October 2016

Figure 12 (framed in blue in Figure 11) shows the new data (November 2015–October 2016), which completes three years of seasonal field monitoring. Graphs of VWCs and temperature are shown consecutively for each trench, with the precipitation (rain/snow) and air temperature at the bottom.

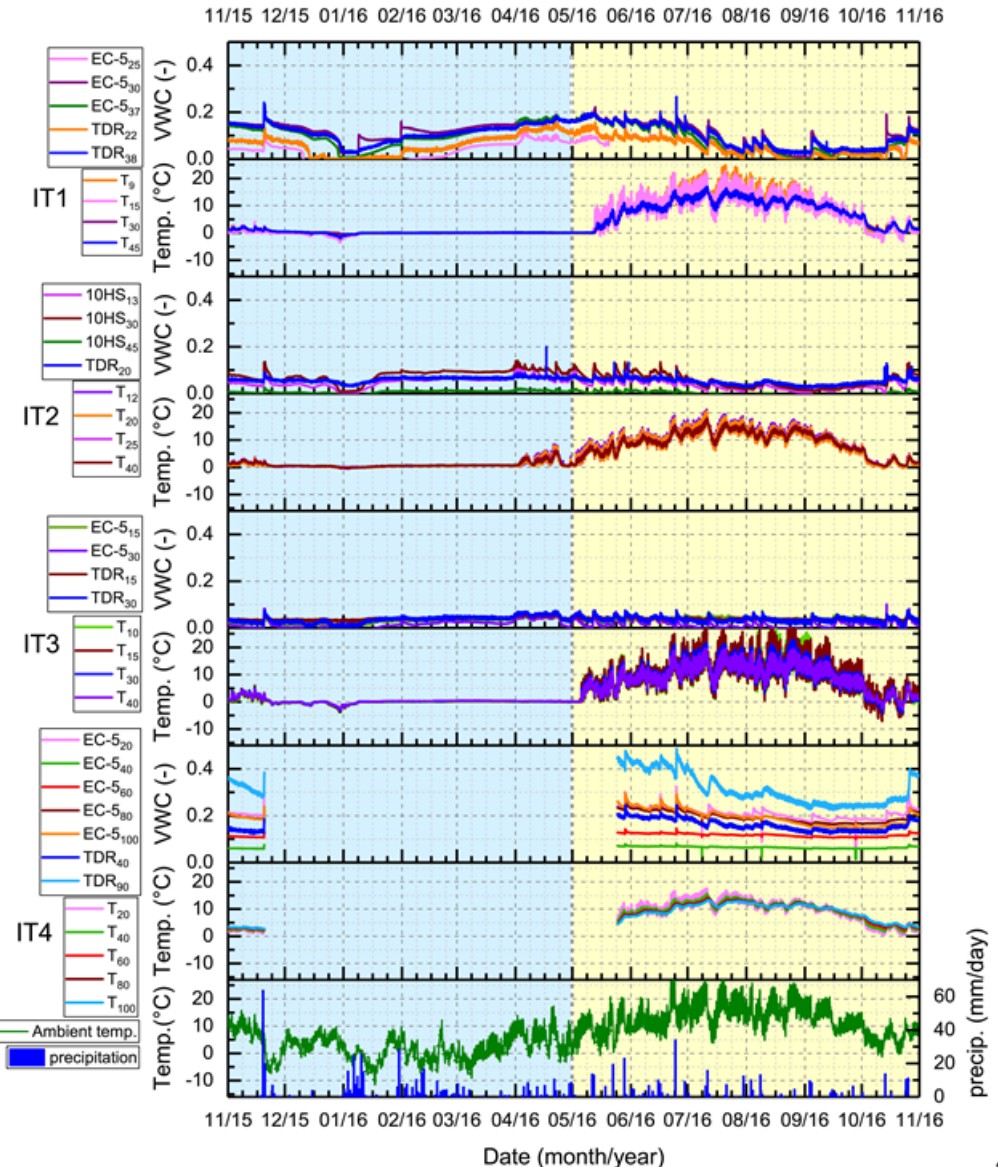

**Figure 12.** Measurements at instrumented trenches IT1-4 during the last year of seasonal field monitoring (November 2015–October 2016): VWC, temperature with precipitation and ambient air data on the bottom.

The new findings of the last year of calibration, in respect of the previous two years, are summarised as follows.

Increment in VWC during Winter Season

A continuous minimum VWC was not observed in any IT during this winter period, and a sustained increased of VWC (January to March 2016) was noticeable in IT1-2. This behaviour of the VWC was slightly observed before during the winter 2015, January (Figure 11, IT1-2), where during a

short period the soil moisture noticeable increased after a brief period of variation between freezing temperatures and warmer temperatures.

The last winter of seasonal monitoring 2016, showed periods of unusually high ambient air temperature, which were predominantly positive (green trace), alternating with intense snowfall (peak of over 60 mm/day after mid-January) and avalanche events (recorded by in situ cameras, which caused loss of data in IT4 from 21 November 2015 to 24 May 2016).

Despite the soil temperature in all of the trenches being recorded as near or below 0 °C, this "short winter" combined with periods of warmer temperatures may have induced early snowmelt, removing the insulating snow layer, facilitating water infiltration and potentially leading to surface erosion.

In addition to the external environmental factors, IT1 is the location with more vegetation than any of the other instrumented trenches followed for isolated patches of vegetation near IT2 and almost nonexistent vegetation around IT3 due to the active erosional processes. Most of vegetation in the slope become dormant in winter due to the extreme weather conditions and layers of snow observed up to 1 m high in one of the cameras near IT2-3. Therefore, no evapotranspiration or increase in suction will be contributed during winter time, enhancing potential increasing of VWC during this season [59]. This is very different to the summer behavior since the evapotranspiration of the plants plus the rainfall and soil type become active factors in the dynamic of the soil water balance.

Erosional events like avalanches in a steeped slope, and rockfalls produce forces able to destroy the existent vegetation (field observations) which could contribute to the generation of a new different scenario from consecutive seasons. Survival, new and removed vegetation could affect the soil characteristics [59,60] in terms of void ratio and range of measured VWC.

VWC Measurements up to 1 m Depth

Due to instability of the soil during excavation in trenches IT1-IT3 their maximum depth reached up to half meter, only IT4 recorded data at greater depths up to 1 m. This trench was located near IT1 (Figures 1, 10 and 11) and had also vegetation and more content of fines (Table 5).

Although the data collection was affected by the avalanche at the beginning of the last winter in 2016, the VWC recorded at IT4 during the summer season provided useful information, especially at deeper locations (up to 1 m), showing higher VWC with depth, and with peaks close to 0.5 in June, indicating that the soil was near to saturation (1 m depth).

3.2.3. Site-Specific Calibration

VWC Measurements over a Range of Temperatures

The VWC sensors used in each trench were recalibrated in soil excavated from each IT (Figures 4, 7 and 13–16), which had been reconstituted at the field relative density (Table 4) with different VWCs (moisture content of 0%, 1%, 3%, 5%, 8% or up to saturation) and temperatures applied, varying from −5 °C to room temperature (17°C to 23 °C). A relationship was determined to calculate the VWC as a function of temperature (Supplementary Material, Figures S7–S14).

Sensor Measurements at Temperatures under 0 °C

In the case of an unfrozen soil, the VWC reflects the combined permittivity of air, ice and unfrozen water [66–69]. Data recorded in the laboratory at negative temperatures were excluded from the recalibration, since the VWC associated to the unfrozen water present in the soil was not measured. The measurements were low similar to those obtained from the original calibration with a range of VWC between 0–0.07 [6].

Impact of Maximum Grain Size

A sensor calibration was completed with a maximum grain size passing the 45 mm and 4mm sieve, using a capacitance sensor EC-5 (at approximately 20 °C) and in soil from IT1 sieved to the GSD

(Figure 7). It shows that the VWC is underestimated at finer gradings, confirming a very significant effect of the GSD on the calibration factors (Supplementary Material Figure S8a).

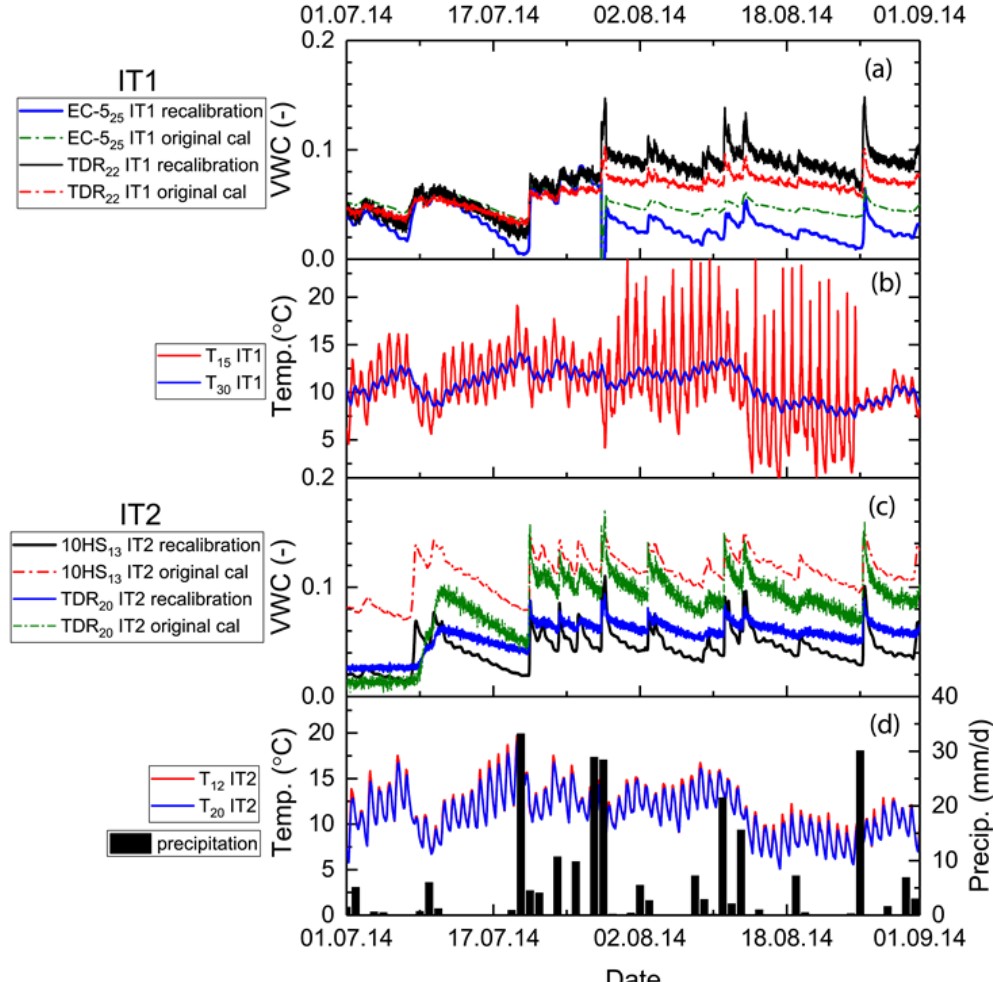

**Figure 13.** Data Period 1: 1 July 2014–31 August 2014. Site-specific calibration: comparison between data based on original calibration and the recalibration for the summer period.

### 3.2.4. Comparisons for Seasonal Responses of the Scree Slope

The VWC and temperature measurements by season are shown in Figures 13–16 with a VWC determined using the original calibration [6] in soil from IT1, and compared to the site-specific recalibration results (2018), performed with soil from each destination trench at an equivalent void ratio.

Summer

Figures 13 and 14 represent the summers of 2014 and 2015 respectively. Each colour represents the VWC measured with either a EC-5/10HS or a TDR sensor at a certain depth, using the original calibration, or the new site-specific recalibration.

Good agreement can be observed between the VWC measured with EC-5 in IT1 in Figures 13a and 14a, since the original calibration and the re-calibration were performed with the same soil. A shift in the VWC in Figure 13a can be observed after a debris flow event (beginning of August), which could have happened due to a changed the local soil conditions in the volume of measurement or sensor performance.

The VWC calculated from the original calibration factors and those from the recalibration show more differentiation in IT2 and IT4 (Figures 13c and 14c,e respectively), due probably to the different

soils, IT1 for original calibration and from the respective trench in the recalibration, and void ratios adopted to represent field conditions in the calibration process. Sensors 10HS$_{13}$ and TDR$_{20}$ in IT2 Figure 13c, appeared to agree better after recalibration, exhibiting notably lower values in VWC (0.05–0.1).

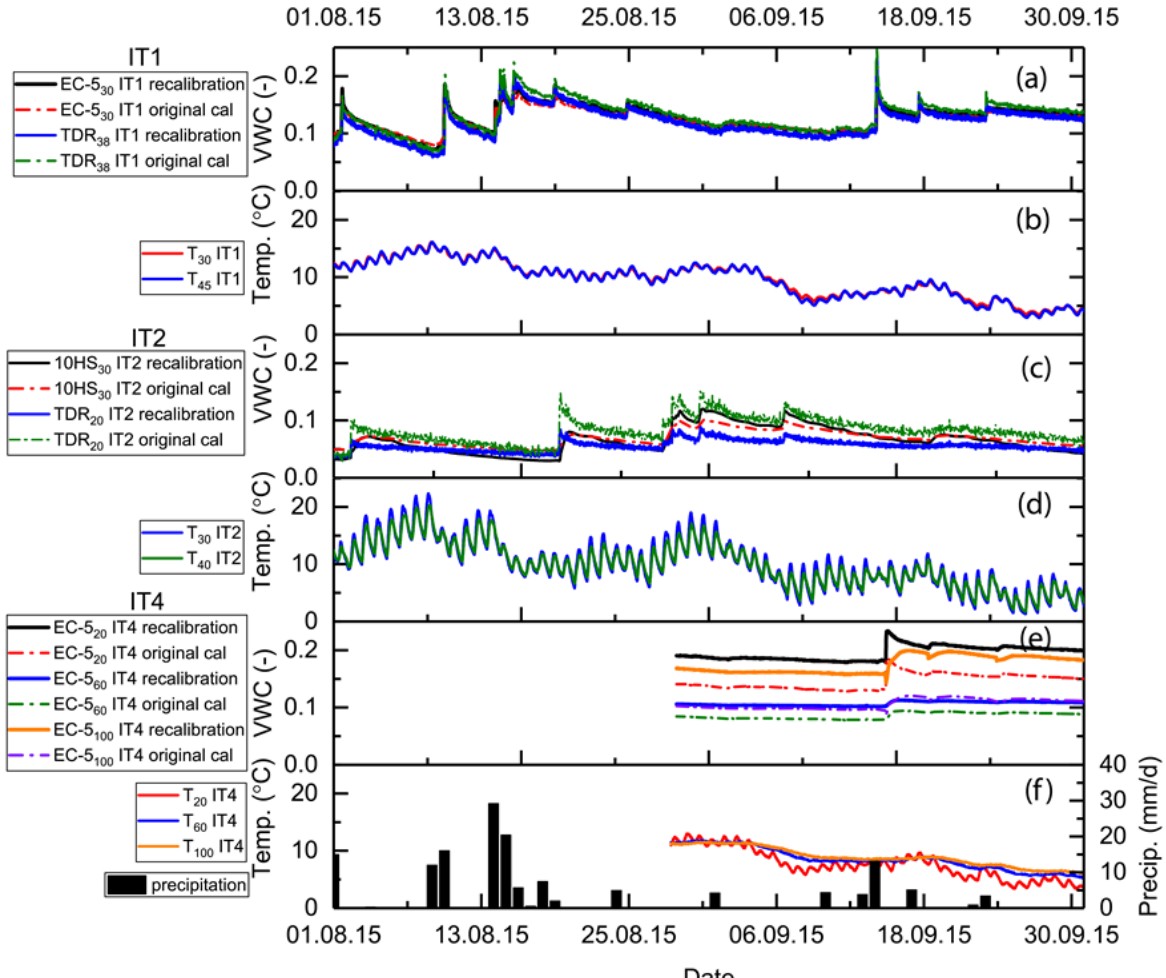

**Figure 14.** Data Period 4: 1 August 2015–30 September 2015. Site-specific calibration: comparison between data based on original calibration and the recalibration for the summer period.

Winter

Figure 15a,c shows a comparison of VWC determined using the original and recalibration equations for the winter period for IT1-2 respectively. A larger difference in the magnitude of VWC in EC-5$_{25}$ Figure 15a was observed after mid-December with negative values, which is not feasible (same sensor with shifted readings after the debris flow (Figure 13a). Furthermore, the soil voids contain air, ice and unfrozen water during the winter regime. The replacement of air by ice in the voids should increase the combined dielectric constant, and therefore VWC should be greater than dry soil with air ($K_{air} = 1.0$; $K_{water} = 78.5$ at 25 °C; $K_{ice} = 3.2$ and $K_{soil} = 3$–7). This suggests that a polynomial calibration equation where the curve flattens at lower VWC, (as shown in Supplementary Material Figure S8e) would be a better fit for the determination of VWC during a winter regime. Some other differences between the original calibrations and the recalibration in IT2 (Figure 15c) could be attributable to the different soils and void ratios. The TDR$_{20}$ Figure 15c shows a greater range of magnitude in the VWC measurement than the 10HS$_{30}$. Finally, peaks in VWC in winter were observed due to precipitation of snow/rain combined with periods of ambient temperatures in a range of (−10 °C to 12 °C).

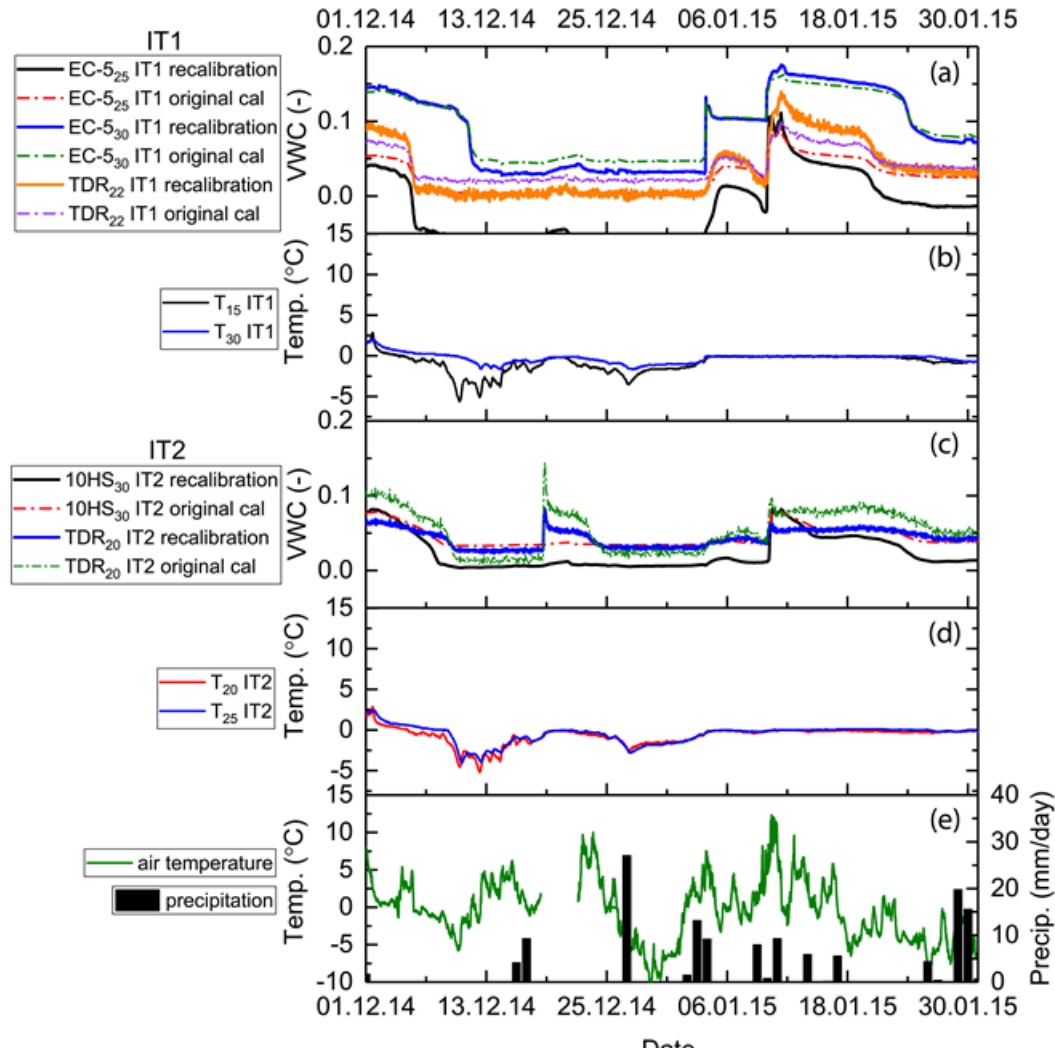

**Figure 15.** Data Period 2: 1 December 2014–30 January 2015. Site-specific calibration: comparison between data based on original calibration and the recalibration for the winter period.

Spring/Autumn

The variations in results for both calibrations are rather insignificant (in engineering terms) in Figure 16a for IT1, EC-5. The TDR data after recalibration displays a greater range in magnitude. IT2 10HS calibrations are similar, with some differences for 10HS$_{45}$ (Figure 16c).

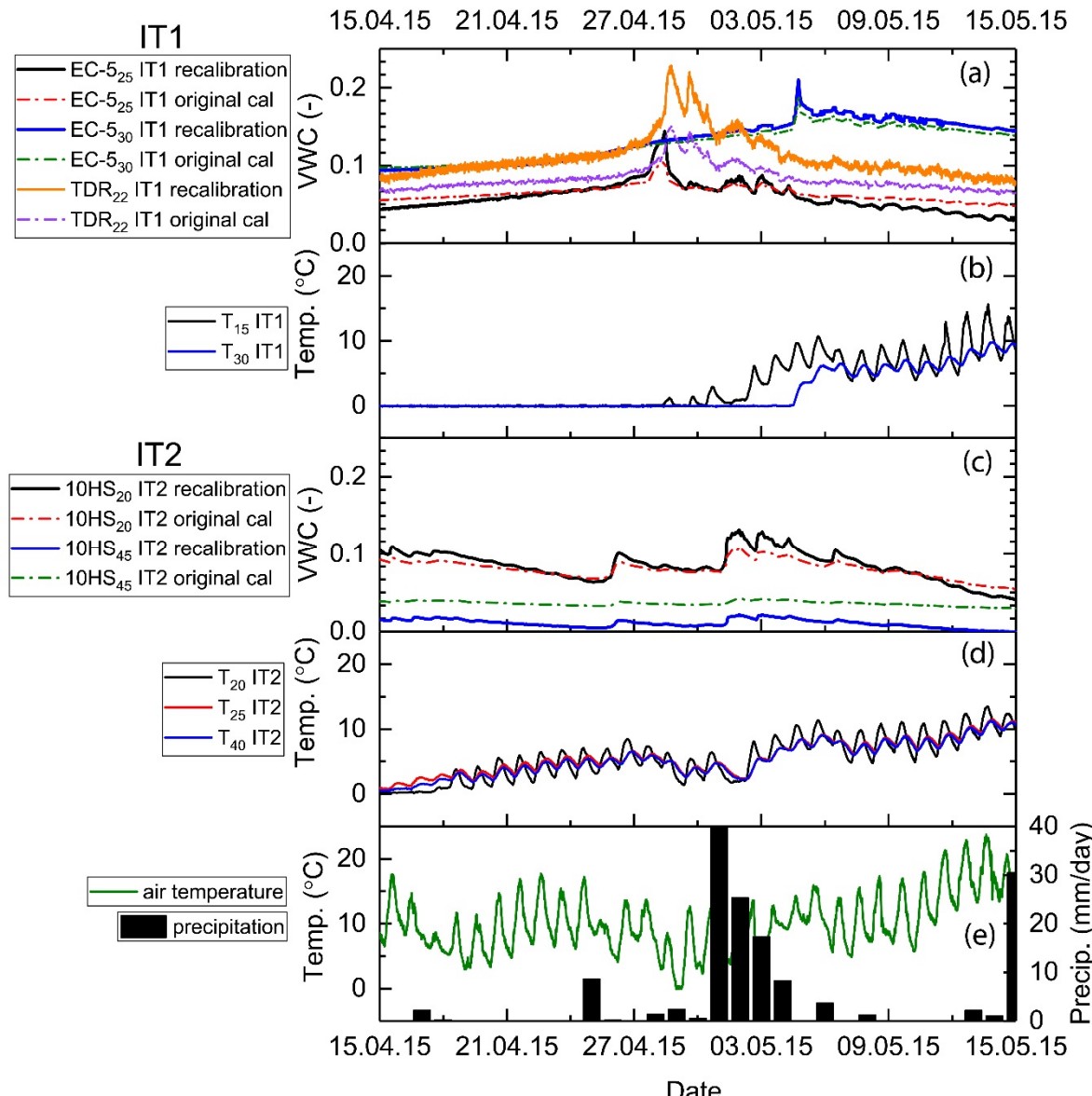

**Figure 16.** Data Period 3: 15 April 2015–15 May 2015. Site-specific calibration: comparison between data based on original calibration and the recalibration for the spring period.

### 3.3. Numerical Modelling

Preliminary simulations of the instabilities caused by saturation of the ground through flow downslope were analysed using four different ground model geometries. Each of these four models represent a scenario of the scree slope (Figure 17), with (a) slope parallel to bedrock with toe, (b) slope parallel to bedrock with no toe (c) bedrock step with toe and (d) bedrock step with no toe. A phreatic water table was achieved by applying an input flow $Q$ at the top of the slope to cause an instability with a F.S. = 1. The flow $Q$ was determined for comparison of the hazard scenarios and given in Table 8.

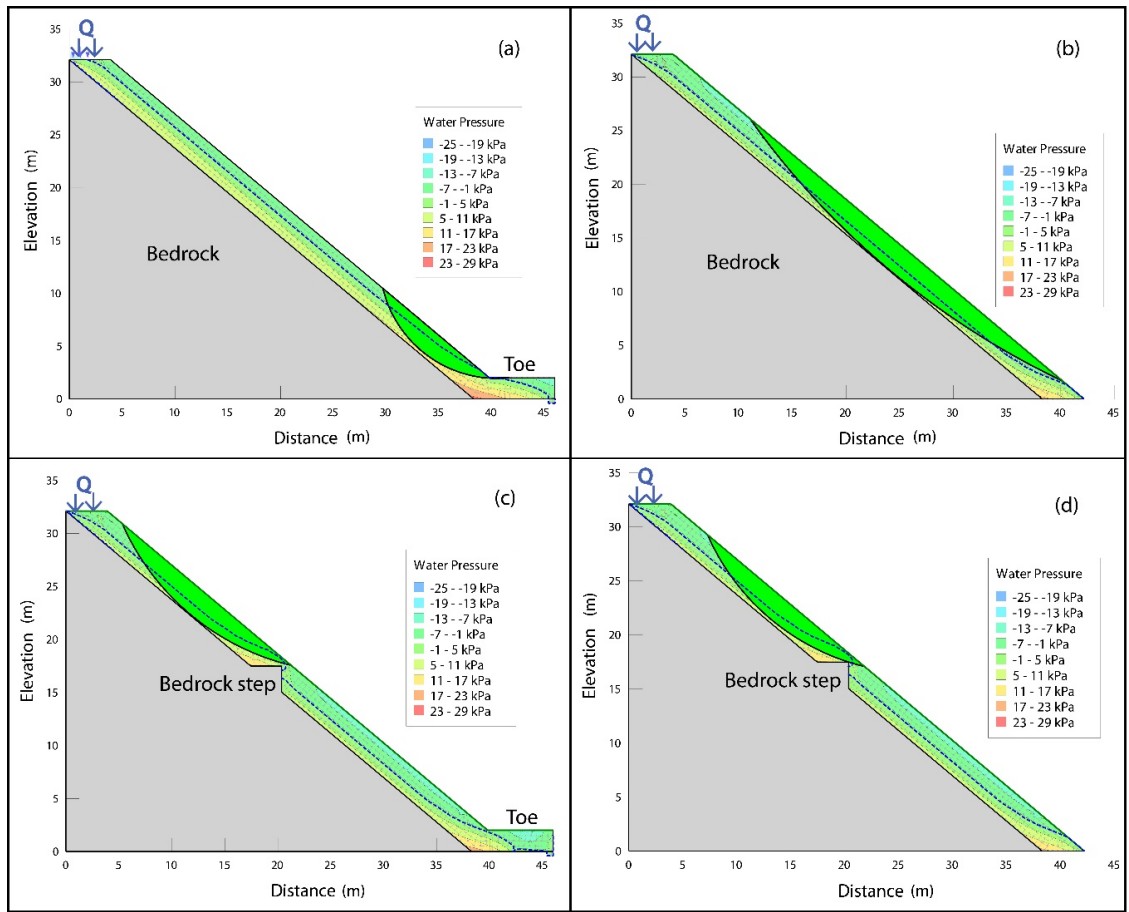

**Figure 17.** Preliminary numerical modelling of instabilities for four different slope bedrock geometries.

**Table 8.** Water flow input to cause slope instability (F.S. = 1) in a numerical model (Geoslope).

| Case | $Q$ (mL/min/m) | Volume Estimated (m$^3$/m) |
|---|---|---|
| (a) slope parallel to bedrock with toe | 819.0 | 19.7 |
| (b) slope parallel to bedrock no toe | 585.0 | 56.6 |
| (c) bedrock step with toe | 562.0 | 34.0 |
| (d) bedrock step no toe | 515.0 | 28.3 |

A slip failure surface is shown in green with the water table plotted as a dashed blue curve. The results show that the safest configuration under the imposed hydraulic conditions would be (a) in that the flowrate at failure is significantly greater and the initial volume displaced is the smallest. A bedrock step case (d) requires the least water flow to fail, and is therefore the most hazardous although the differences (c) and (d) are small, because they are affected primarily by the bedrock step. The role of the toe is important in that it prevents the water from emerging as a spring on the surface, leading to development of an initial local failure mechanism. The volume mobilised was greatest in case (b).

## 4. Interpretation, Discussion and Implications for Practice

### 4.1. Characterisation

#### 4.1.1. Grain Size Distribution (GSD)

The GSD at IT1-4 was measured at the beginning (2014) and at the end of the seasonal field monitoring (2016) with the same sampling techniques. Similar results are obtained ([6]; Figure 7). Although dynamic processes are affecting the scree slope, the composition of the gravelly soil did

not show significant variations at these specific locations during the three years of seasonal field monitoring, due to the self-sorting and gravitational processes. This confirms the observations from the in situ cameras that while surface movements and ravelling occurred, no massive mass movement event has taken place in these locations in recent memory.

### 4.1.2. Unit Weight

The in-situ unit weight measurement with the balloon method was challenging in the gravelly scree soil. Problems happened during excavation due to the heterogeneity and the larger grain sizes and because the angular gravel could damage the rubber balloon. Improved experience of the user allowed the technique to be applied more successfully after repeated measurements. The results in Table 4 could be grouped into downslope trenches (IT1–IT4) with a higher dry unit weight >19 kN/m$^3$, and upslope trenches (IT2, IT3), with values around 18 kN/m$^3$. This natural segmentation could be explained by the sorting processes along the scree slope. Larger stones move greater distances downslope than smaller particles [2], whereas suffusion [70] may cause finer sands and silts to be transported downslope by seepage, as observed in the GSD curves of IT1 and IT4 (Figure 7). This was corroborated during field observation of loose blocks upslope and soil inspection in the trenches.

### 4.1.3. Triaxial Testing

The determination of soil strength parameters for the gravel was successfully performed by using a medium size and large size diameter triaxial devices and a stress path that simulates slope failure due to rainfall. The critical state friction angle was shown to be similar in both cases (41° and 42° respectively). Some relevant aspects of the testing were:

- Soil specimens were prepared successfully to loose to medium density by moist tamping with 3% moisture content (Table 2), based on an average of the GSD from IT1-4 (Figure 5b). The hydraulic gradient (around 1.0) applied during saturation (from bottom to top) could have favoured transportation of the fines (suffusion). This could explain the volume loss during saturation (with vacuum and $CO_2$ methods), which was significant in the mid-size specimen, although this mechanism could not be confirmed.

- Considering that soil in steep slopes experiences a highly anisotropic stress state [35,38,41], the tests were carried out with a $K_c$ = 1.83 to 2.42 (Table 2), which was still conservatively lower than $K_c$ = 3.3–4.3, calculated as one tenth of the slope angle (33°–43°), (as used by Anderson [34] in a residual and colluvial soil).

- The CSD triaxial stress path reproduced the mechanism of failure due to rain infiltration, increasing the PWP and moving the stress path horizontally with decreasing p′ in a $q$-$p'$ space to intersect the failure envelope (stress ratio $K_F$), this would cause soil elements to fail and could lead to a landslide.

- The gravel exhibited dilatant behaviour during shearing for both the medium and large triaxial specimens. Although the specimens were constructed at a loose-medium relative density ($e$ = 0.49–0.59), they densified during the saturation and consolidation stages to void ratios of $e$ = 0.27–0.58. Particle breakage was confirmed during one of the large triaxial tests (Supplementary Material, Figure S3, Table S2) at the highest confining pressure of ($\sigma'_{1,B}$ = 227.4; $\sigma'_{3,B}$ = 94.1 kPa) by conducting GSD tests before making the specimen and after shearing.

- A unique critical state friction angle of 41° (no cohesion) was determined from tests in the medium size triaxial apparatus, whereas 42° was determined in the large triaxial, also with no cohesion. Dilatancy was enhanced at lower confining pressures because of the effect of particle interlocking and the GSD.

- It would be recommended for the mid-scale triaxial apparatus to run tests at a higher confining pressure, to obtain the critical state strength parameters and to improve determination of the CSL, and to observe the dependency of the soil behaviour and strength parameters on the confining

pressure. This task will additionally allow the comparison of the CSL obtained from mid and large-scale specimens at a similar range of effective stresses, and check on the dilatancy of the adopted GSD.

### 4.1.4. Ground Penetrating Radar (GPR)

The GPR measurements were crucial for the soil characterisation and the definition of a ground model. The profiles set up were according to, and representative of, the scale of the scree slope covering the whole area of study. Reliable results were obtained. A contour map showing the depth to bedrock was determined for selected criteria, and this was critical for the hazard assessment.

### *4.2. Seasonal Field Monitoring*

Two years of seasonal field monitoring have been reported in [6]. A third year of data is added to finalise the monitoring campaign and a new site recalibration was performed using the soil and void ratio corresponding to each instrumented trench in the field.

The main aspects to discuss about the seasonal field monitoring include the additional year of data and the site-specific recalibration of the sensors:

### 4.2.1. Additional Year of Seasonal Field Monitoring

The last year of data included a winter and a summer period (2015 to 2016) and confirmed the trend of VWC and temperature measured in the soil during the entire three year period. Variations and ranges in VWC and temperature in the soil for each trench depended not only on the soil characteristics, but were particularly responsive to changes in climatic conditions (rain (summer), snow-melt and rain (spring/autumn), and snow fall combined with higher ambient temperatures (winter)) and location within the scree slope (inclination, sun exposure).

The addition of data at IT4 of up to and including 1 m depth was also valuable, since it showed a higher and wider range of VWC than the rest of the trenches, with peaks of saturation at greater depths (Figure 12), but drainage was fast due to the high permeability, similar to its neighbour IT1 (up to 0.5 m depth). The soil did not reach saturation at any season in the rest of the trenches during the entire three year period.

Although this research focused on the response of VWC to water infiltration, the VWC behaviour in winter-time is of interest, especially in an alpine environment, due to the complex interaction between air, ice, an insulating snow layer, unfrozen water, and infiltrating water. Furthermore, a study of the freezing and thawing during winter and their combined effects on the water infiltration, run off and erosion could lead to a deeper understanding of the hazard associated to snow melting and avalanches, as well as their influence in the occurrence of shallow landslides in the scree slope.

The effect of the vegetation in the variation of VWC in the case of an alpine scree slope, where climate and hazards are a constant challenge would be of an addition to a future similar study.

### 4.2.2. Site-Specific Recalibration Sensors for Determination of VWC

A site-specific recalibration was performed under laboratory-controlled conditions to determine the influence of the temperature variations in gravelly soil over a range of temperature of −6 °C–23 °C. Some of the challenges and findings were summarised in the Supplementary Material.

The effect of temperature on the determination of VWC was of special interest because of the variation of temperatures at the field site. The influence of temperature on the dielectric constant of the soil has been mentioned by other authors [52–56]. They present different evidence-based opinions on whether there are changes in the dielectric constant and whether these lead to an increase or decrease, although not much information about gravels was found in the literature. The site-specific calibration of sensors at the Meretschibach included variations in temperature, ranging from −6 °C to 23 °C. The results show small variations that could be attributed to the heterogeneity of the soil, and so the relationship between the soil temperatures and the estimation of the VWC was assumed to be

related primarily to the grading of the soil in situ, locally, around the sensors. Hence, it was found that a site-specific calibration is highly recommended when monitoring gravelly soil ($D_{max}$ = 31.5 mm), because the use of smaller sizes (e.g., $D_{max}$ = 4 mm of the same soil) could lead to an underestimation of the VWC.

### 4.3. Numerical Modelling

As hypothesised, a berm is demonstrably helpful in managing the emergence of water outflow onto the slope without a bedrock step, whereas the presence of steps in the bedrock makes the situation more critical. The bedrock step forces the PWP to rise at the junction of the step and the bedrock slope, saturating the soil upslope, and reducing the effective stresses. If the bedrock step was close to the surface, water could outflow there, in a form of a spring, eroding the soil at this location and triggering a failure (Figure 17c,d). The toe favoured an increase of the PWP and failure in the bottom section (Figure 17a), whereas a larger volume of debris could be mobilised in a slope with no toe and bedrock step (Figure 17b, Table 8).

It would be recommended to carry out further numerical modelling in the future, when such events have taken place and to validate these with physical models that are able to represent the field conditions and features of the mechanism of failure, in terms of pore water pressure development, potential cracking, and run out.

### 4.4. Hazard

The results in this study focused on one of the possible hazards, specifically surficial landslides, in the scree slope. The relevance of this type of hazard is that it could change the stability of the scree slope on a larger scale, and could also influence the occurrence of other hazards such as debris flow. It is possible that the risks of rainfall-induced landslides could be studied by adopting a quantitative risk assessment framework, for example as proposed by [71], by engaging in deterministic and probabilistic analysis of the failure occurrence for rainfall-induced landslides (e.g., [72,73])

In terms of slope stability assessment and based on the results, the locations of the scree slope, which are more endangered, are locations where:

- the bedrock outcrop approaches the ground surface (relatively shallow depths to bedrock (e.g., 1–1.5 m) (Figure 10). In this case, the groundwater flow can saturate the soil layer and potentially form a spring at the surface, causing the soil to erode and fail;
- the slope location is more susceptible to erosion processes from snow melting and rainfall run off, which that can contribute to the remobilisation of debris in the slope.
- It would be recommended to measure the potential water table (if there is any) depth in order to analyse further the hazard. An instrumented well near IT4 (due to the relatively flat ground surface and the less coarse grain sizes, which facilitate the drilling) would meet the purpose

## 5. Summary and Conclusions

A geotechnical characterisation of an alpine scree slope at the Swiss Alps in Switzerland, was conducted to provide the basis for the hazard assessment of shallow landslides triggered by rainfall combined with an antecedent of high groundwater tables. Work reported in a previous paper [6] was extended with additional data and key information including soil strength parameters, a map of the depth to bedrock, the long-term seasonal field monitoring data, and a preliminary numerical analysis of the slope stability for four different cases of bedrock-slope geometry.

The Böchtur slope, of predominantly steepness (33°–43°), is in a state of incipient failure for surficial erosion and at risk of shallow slips, given critical state friction angles of 42° and 41° respectively and zero cohesion. These parameters were determined for a representative soil grading through triaxial stress path tests on large specimens (up to 0.25 m diameter), and representing shearing during slope failure under rainfall infiltration. Low confining stresses near to the surface might lead to some

enhanced strength due to dilatancy associated with the interlocking of the coarser gravel. Locally, vegetation reinforces the surface through root growth, however the heterogeneity and sorting in the soil deposits also mean that some areas of the slope are more at risk of surficial failure than others. This process is dynamic and the likelihood of failure will decrease with increasing depth of soil to bedrock and increase with a stepped geometry of the bedrock and saturation of the ground.

The maximum volume of soil debris overlying bedrock is of the order of 15,000 m$^3$; however, mobilising the entire volume would require full saturation of a significant proportion of the overlying soil layer, which was not observed during the three years of seasonal field monitoring. The amount of snowmelt, rainfall infiltration and groundwater flow that would be necessary to achieve this state would require a highly unusual combination of deep snow layers, late spring melt and highly extreme rainfall events. Localisation of where the bedrock is shallower and therefore where the overlying soil layer is more likely to reach saturation will influence where smaller volumes of debris might be released. Numerical modelling simulations can quantify the amount of groundwater flow required to cause failure, dependent upon soil properties, soil depth and bedrock geometry. Typically, these conditions will influence the location of the failure and the debris volume, which will be limited by and happen upslope of a bedrock step. Otherwise any failure might be expected to initiate from the bottom a slope if the bedrock and slope are parallel to each other.

**Supplementary Materials:** The following are available online at http://www.mdpi.com/2073-4441/12/2/447/s1, Figure S1: Balloon method device used in in situ soil unit weight measurements in gravelly soil at the scree slope, Figure S2: Triaxial testing apparatuses, Figure S3: Grain size distribution before testing and after shearing for large scale test 3, Figure S4: Soil in reconstituted samples for calibrating sensors from IT1-4, Figure S5: Sensors for each of the instrumented trenches IT1-4, Figure S6: Controlled room temperature at IGT ETH Zürich, Figure S7: Site-specific calibration of VWC sensors of IT1, IT2, IT4 and the effect of temperature, Figure S8: Site-specific calibration for IT1, IT2 and IT4. For a range of temperature (1 °C to 23 °C), Figure S9: Site-specific calibration of TDR and EC-5 sensors in terms of temperature IT1, Figure S10: Site-specific calibration of TDR and 10HS sensors in terms of temperature, IT2, Figure S11: Site-specific calibration of TDR and EC-5 sensors in terms of temperature, IT4, Figure S12: Site specific calibration for positive temperatures, IT1, Figure S13: Site-specific calibration for positive temperatures, IT2, Figure S14: Site-specific calibration for positive temperatures, IT4, Table S1: Specifications of the triaxial apparatuses, Table S2:Diameter $D_i$ for large triaxial specimen before and after shearing, Table S3: Overview of all acquisition parameters used during GPR surveys. "PE" stands for PulseEKKO, and "GB" for ground-based acquisition, Table S4: Selected processing parameters for each GPR acquisition.

**Author Contributions:** The first author was D.L., who wrote 75% of the paper and prepared 90% of the figures and all of the tables. She performed the seasonal field monitoring over a three years period, carried out laboratory testing including 15%–20% of the triaxial testing, all of the grain size distributions and site-specific recalibration of volumetric water content sensors, data analysis and processing, and she carried out the preliminary numerical analysis. She also contributed to the hazard assessment. K.F. performed the geophysical measurements (GPR), and prepared the related figures (Figure 1a,b, Figure 9. GPR profile and Figure 10 Contour Map) and contributed with the text and discussion for the geophysical results with H.M., who also contributed to writing the geophysics section, reviewing, guiding and supervising during the measurements, the planning of the paper, discussion on the hazard assessment, and review of the overall paper. B.M. contributed to writing the hazard assessment and the related Figures 2 and 3; and review of the overall paper. R.G. built on his master's project, performed 5 of the 6 triaxial tests under the first author's supervision and carried out the first analysis of the stress-path test data. R.H. contributed with field assistance during construction of the instrumented trenches, soil sampling and supervision on the laboratory triaxial testing. In discussion with D.L., he also installed a meteostation. E.B. helped in the installation of sensors and data collection in the field site (3 years period seasonal field monitoring). S.M.S. initiated the project and provided direct supervision, guidance on the structure and during writing of the paper, as well as comprehensive review and feedback. She wrote the conclusions. All authors have read and agreed to the published version of the manuscript.

**Funding:** The authors are most grateful for funding from the SNF Project No. 200021_144326/1 as well as supplementary support from Canton Valais, together with the Councils of Agarn and Leuk.

**Acknowledgments:** The authors express their gratitude to the Councils of Leuk and Agarn, the local technicians, who collaborated during the seasonal field monitoring, to Nicole Oggier for her help during the first stages of the monitoring, and to the IGT technicians at the workshop for the tools created for the monitoring. The authors also wish to acknowledge assistance from Thomas Buchli, Andrew Kos and Alexandru Marin, for their collaboration in the fieldwork tests and tasks, to Amin Askarinejad for his instruction and feedback on instrumentation, as well as Andre Nuber, Fabienne Reise, Marlies Vasmel and Melchior Grab for their help in the acquisition of the GPR data.

**Conflicts of Interest:** The authors declare no conflict of interest.

## List of Notation

| | |
|---|---|
| CSD | Constant shear stress drained path |
| CSL | Critical state line |
| CADCAL | Anisotropic consolidation-drained shear at constant axial load |
| ERT | Electrical resistivity tomography |
| EC-5 | Capacitance sensor, Decagon Devices |
| F.S. | Factor of Safety |
| GSD | Grain size distribution |
| GPS | Global positioning system |
| GPR | Ground penetrating radar |
| ID | Intensity-duration threshold |
| IGT | Institute for geotechnical engineering |
| inSAR | Interferometric Synthetic Aperture Radar |
| IT | Instrumented Trench |
| m.a.s.l. | Metres above sea level |
| PWP | Pore water pressure |
| TDR | Time domain reflectometry sensor, Campbell Scientific |
| VWC | Volumetric water content |
| WSL | Swiss Federal Institute for Forest, Snow and Landscape Research |
| $C_c$ | Coefficient of curvature |
| $C_u$ | Coefficient of uniformity |
| $c'$ | Cohesion |
| $D_i$ | i% of the particles are finer than this size |
| $D_{r,i}$ | Initial relative density (before consolidation) |
| $e$ | Void ratio |
| $\varepsilon_a$ | Axial strain |
| $\varepsilon_v$ | Volumetric strain |
| $K_c$ | Principal stress state ratio |
| $K_F$ | Principal stress state ratio at failure |
| $K_{water/ice/soil}$ | Dielectric constant of (water/ice/soil) |
| $p'$ | Mean effective stress |
| $q$ | Deviatoric shear stress |
| $Q$ | Input water Flow |
| 10HS | Capacitance sensor, Decagon Devices |
| $\phi'$ | Critical state friction angle |
| $Y_d$ | Dry unit weight |
| $\psi$ | Dilatancy angle |
| $\sigma'_i$ | Principal effective stress |
| $q_A, p'_A$ | Mean effective and deviatoric stress after isotropic consolidation |
| $q_B, p'_B$ | Mean effective and deviatoric stress after anisotropic consolidation |
| $q_C, p'_C$ | Mean effective and deviatoric stress at failure |
| $\sigma'_{1,B}, \sigma'_{3,B}$ | Principal effective stresses 1, 3 before shearing |
| $\sigma'_{1,C}, \sigma'_{3,C}$ | Principal effective stresses 1, 3 at failure |
| $k$ | hydraulic conductivity (m/s) |
| GP-GM | Poorly graded gravel with silt and sand |
| GP | Poorly graded gravel |
| GM | Silty gravel |
| $\varepsilon_{a,C}$ | Axial strain at failure |
| $\varepsilon_{v,C}$ | Volumetric strain at failure |
| $M$ | Slope of the CSL |
| $e_0$ | Initial void ratio (before consolidation) |
| $e_B$ | Void ratio before shearing |
| $e_C$ | Void ratio at failure |
| $I_{D,0}$ | Initial relative density (-) |
| $I_{D,C}$ | Relative density at failure (-) |
| $I_R$ | Dilatancy Index |

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
