# Peer review of "Slope Stability of a Scree Slope Based on Integrated Characterisation and Monitoring"

_water, doi:10.3390/w12020447_

Round 1

Reviewer 1 Report

This paper investigates the mechanisms of a surficial failure in the scree slope under rainfall and different bedrock conditions. Overall, this work has some scientific contributions and provides a lot of valuable field data, which are beneficial for other researchers and engineers in the related areas. The research tools are very comprehensive, including field monitoring, laboratory testing and numerical modelling. There are some comments/suggestions that should be addressed before the paper can be considered for publications. Major revisions are suggested.

The structure of the manuscript needs to be reorganised. For example, Section 2 “The Meretschibach-Böchtur scree slope experiment” should be incorporated into Section 3 “Methodology”. ‘4.1.1 Dry unit weight” and “4.1.2 Grain size distribution” should be moved to “3.1.1 Soil Unit weight and grain size distribution”. This paper adds one more year of data to that already published (i.e., two years). The authors should highlight the new findings from this one more year of data when compared with the previous two years. The sensor calibration results should be moved to appendix. Those calibration results do not tell anything new. It is mentioned in L105-106 that” stunted vegetation, like spruce and low shrub of less than 1 m height, are only partially present along the slope…”. The authors are highly encouraged to write few sentences to discuss the hydro-mechanical effects of vegetation on slope stability. You can refer the latest work, such as “Modelling hydro-mechanical reinforcement of vegetation to slope stability” and “Mechanical and hydrological impacts of tree removal on a clay fill railway embankment.” Quality of some figures/photos should be improved, such as Figure 2 and 10. Boundary conditions should be labelled in the model geometry in Figure 6. Why is Figure 9 put in the front? The lines in all figures should be drawn in different line types. They are now all solid lines with different colours. When printed in black and white, they can not be differentiated. The authors are highly recommended to compare the numerical results with the field results. Otherwise, the simulation is just like a mathematical game, without any true physical meaning. When using SEEP/W, how are the input soil hydraulic parameters (i.e., soil water retention and soil permeability function) determined? “5. Interpretation and discussion”: this section should be enhanced by providing more insights. The conclusions are too long. Authors should just highlight the new findings only in the conclusions.

Reviewer 2 Report

This is a very data heavy/dense paper.  Overall it is well written.  The authors need to revise the paper however to improve its formatting especially its frequent use of sub-sections/sub-sub sections.

I will also ask the authors to review and ensure all figures are necessary, and if so, if they are all necessary in color.
Specific comments are given below
AbstractGPR has not been defined before use in line 19recommend adding "groundwater" before "flow" in line 25
Figure 1 needs to be greatly improved.  It looks like someone used a marker to scratch off part of a title.

Figure 9 must be misplaced (page 3?) where are figures 2-8?
Figure 9 caption should be revised.  Is  "ground-based" necessary when we are talking about ground penetrating radar (GPR)?
The authors should spell out all acronyms in figure and table captions.  Recall, all figures and tables must be able to stand on their own and not require the reader to search through the body of text to figure out an acronym.
Line 206, 253 etc.Should "Soil Shear Strength Parameters", "Soil Specimen Preparation" and "Testing Program" be sub sections.  If so, they should be prefixed with 3.1.2.1 or similar according to the journals guidelines.
(see line 385, 397, 404 also for formatting)
Line 497, this paragraph will ready better as:"VWC and temperature were measured at specific locations of the slope (IT1-4). These data were complemented by precipitation data from two meteostations..."
For all figures (especially noted in fig 11), you should not use both a title and a caption.  Eliminate the title and provide all necessary information to the reader in the caption..eliminate blank line, line 776

Round 2

Reviewer 1 Report

The authors have addressed my concerns satisfactorily. The revised manuscript can be accepted for publication now.